# Data-Dependent Bounds for Online Portfolio Selection Without Lipschitzness and Smoothness

**Chung-En Tsai**
Department of Computer Science and Information Engineering
National Taiwan University
CHUNGENTSAI@NTU.EDU.TW

**Ying-Ting Lin**
Department of Computer Science and Information Engineering
National Taiwan University
R08922060@NTU.EDU.TW

**Yen-Huan Li**
Department of Computer Science and Information Engineering
Department of Mathematics
Center for Quantum Science and Engineering
National Taiwan University
YENHUAN.LI@CSIE.NTU.EDU.TW

## Abstract

This work introduces the first small-loss and gradual-variation regret bounds for online portfolio selection, marking the first instances of data-dependent bounds for online convex optimization with non-Lipschitz, non-smooth losses. The algorithms we propose exhibit sublinear regret rates in the worst cases and achieve logarithmic regrets when the data is "easy," with per-round time almost linear in the number of investment alternatives. The regret bounds are derived using novel smoothness characterizations of the logarithmic loss, a local norm-based analysis of following the regularized leader (FTRL) with self-concordant regularizers, which are not necessarily barriers, and an implicit variant of optimistic FTRL with the log-barrier.

## 1 Introduction

Designing an optimal algorithm for online portfolio selection (OPS), with respect to both regret and computational efficiency, has remained a significant open problem in online convex optimization for over three decades[1]. OPS models long-term investment as a multi-round game between two strategic players—the market and the INVESTOR—thereby avoiding the need for hard-to-verify probabilistic models for the market. In addition to its implications for robust long-term investment, OPS is also a generalization of probability forecasting and universal data compression [5].

The primary challenge in OPS stems from the absence of Lipschitzness and smoothness in the loss functions. Consequently, standard online convex optimization algorithms do not directly apply. For example, standard analyses of online mirror descent (OMD) and following the regularized leader (FTRL) bound the regret by the sum of the norms of the gradients (see, e.g., the lecture notes by

---

[1]Readers are referred to recent papers, such as those by Luo et al. [22], van Erven et al. [35], Mhammedi and Rakhlin [24], Zimmert et al. [38], and Jézéquel et al. [17], for reviews on OPS.

37th Conference on Neural Information Processing Systems (NeurIPS 2023).

Orabona [28] and Hazan [12]). In OPS, the loss functions are not Lipschitz, so these analyses do not yield a sub-linear regret rate. Without Lipschitzness, the self-bounding property of smooth functions enables the derivation of a "small-loss" regret bound for smooth loss functions [33]. However, the loss functions in OPS are not smooth either.

The optimal regret rate of OPS is known to be $O(d \log T)$, where $d$ and $T$ denote the number of investment alternatives and number of rounds, respectively. This optimal rate is achieved by Universal Portfolio [9, 10]. Nevertheless, the current best implementation of Universal Portfolio requires $O(d^4 T^{14})$ per-round time [20], too long for the algorithm to be practical. Subsequent OPS algorithms can be classified into two categories.

- Algorithms in the first category exhibit near-optimal $\tilde{O}(d)$ per-round time and moderate $\tilde{O}(\sqrt{dT})$ regret rates. This category includes the barrier subgradient method [26], Soft-Bayes [30], and LB-OMD [34].
- Algorithms in the second category exhibit much faster $\tilde{O}(\text{poly}(d) \text{polylog}(T))$ regret rates with much longer $\tilde{O}(\text{poly}(d) \text{poly}(T))$ per-round time, which is, however, significantly shorter than the per-round time of Universal Portfolio. This category includes ADA-BARRONS [22], PAE+DONS [24], BISONS [38], and VB-FTRL [17].

The aforementioned results are worst-case and do not reflect how "easy" the data is. For instance, if the price relatives of the investment alternatives remain constant over rounds, then a small regret is expected. *Data-dependent regret bounds* refer to regret bounds that maintain acceptable rates in the worst case and much better rates when the data is easy. In this work, we consider three types of data-dependent bounds.

- A *small-loss bound* bounds the regret by the cumulative loss of the best action in hindsight.
- A *gradual-variation bound* bounds the regret by the gradual variation (6) of certain characteristics of the loss functions, such as the gradients and price relatives.
- A *second-order bound* bounds the regret by the variance or other second-order statistics of certain characteristics of the loss functions, such as the gradients and price relatives.

Few studies have explored data-dependent bounds for OPS. These studies rely on the so-called no-junk-bonds assumption [2], requiring that the price relatives of all investment alternatives are bounded from below by a positive constant across all rounds. Given this assumption, it is easily verified that the losses in OPS become Lipschitz and smooth. Consequently, the result of Orabona et al. [29] implies a small-loss regret bound; Chiang et al. [8] established a gradual-variation bound in the price relatives; and Hazan and Kale [14] proved a second-order bound also in the price relatives. These bounds are logarithmic in the number of rounds $T$ in the worst cases and can be constant when the data is easy.

The no-junk-bonds assumption may not always hold. Hazan and Kale [14] raised the question of whether it is possible to eliminate this assumption. In this work, we take the initial step towards addressing the question. Specifically, we prove Theorem 1.1.

**Theorem 1.1.** *In the absence of the no-junk-bonds assumption, two algorithms exist that possess a gradual-variation bound and a small-loss bound, respectively. Both algorithms attain $O(d \text{polylog}(T))$ regret rates in the best cases and $\tilde{O}(\sqrt{dT})$ regret in the worst cases, with $\tilde{O}(d)$ per-round time.*

Theorem 1.1 represents the first data-dependent bounds for OPS that do not require the no-junk-bonds assumption. To the best of our knowledge, this also marks the first data-dependent bounds for online convex optimization with non-Lipschitz non-smooth losses. In the worst cases, Theorem 1.1 ensures that both algorithms can compete with the OPS algorithms of the first category mentioned above. In the best cases, both algorithms achieve a near-optimal regret with a near-optimal per-round time, surpassing the OPS algorithms of the second category in terms of the computational efficiency. Table 1 in Appendix A presents a detailed summary of existing OPS algorithms in terms of the worst-case regrets, best-case regrets, and per-round time.

We also derived a second-order regret bound by aggregating a variant of optimistic FTRL with different learning rates. The interpretation of the result is not immediately clear. Therefore, we detail the result in Appendix I.

**Technical Contributions.**   The proof of Theorem 1.1 relies on several key technical breakthroughs:

- Theorem 3.2 provides a general regret bound for optimistic FTRL with self-concordant regularizers, which may not be barriers, and time-varying learning rates. The bound generalizes those of Rakhlin and Sridharan [31] and Zimmert et al. [38, Appendix H] and is of independent interest.
- Existing results on small-loss and gradual-variation bounds assume the loss functions are smooth, an assumption that does not hold in OPS. In Section 4, we present Lemma 4.3 and Lemma 4.7, which serve as local-norm counterparts to the Lipschitz gradient condition and the self-bounding property of convex smooth functions, respectively.
- To apply Theorem 3.2, the gradient estimates and iterates in optimistic FTRL must be computed concurrently. Consequently, we introduce Algorithm 2, a variant of optimistic FTRL with the log-barrier and validate its definition and time complexity.
- The gradual-variation and small-loss bounds in Theorem 1.1 are achieved by two novel algorithms, Algorithm 3 and Algorithm 4, respectively. Both are instances of Algorithm 2.

**Notations.**   For any natural number $N$, we denote the set $\{1, \ldots, N\}$ by $[N]$. The sets of non-negative and strictly positive numbers are denoted by $\mathbb{R}_+$ and $\mathbb{R}_{++}$, respectively. The $i$-th entry of a vector $v \in \mathbb{R}^d$ is denoted by $v(i)$. The probability simplex in $\mathbb{R}^d$, the set of entry-wise non-negative vectors of unit $\ell_1$-norm, is denoted by $\Delta_d$. We often omit the subscript for convenience. The closure and relative interior of a set $\mathcal{X}$ is denoted by $\mathrm{cl}\,\mathcal{X}$ and $\mathrm{ri}\,\mathcal{X}$, respectively. The $\ell_p$-norm is denoted by $\|\cdot\|_p$. The all-ones vector is denoted by $e$. For any two vectors $u$ and $v$ in $\mathbb{R}^d$, their entry-wise product and division are denoted by $u \odot v$ and $u \oslash v$, respectively. For time-indexed vectors $a_1, \ldots, a_t \in \mathbb{R}^d$, we denote the sum $a_1 + \cdots + a_t$ by $a_{1:t}$.

## 2   Related Works

### 2.1   Log-Barrier for Online Portfolio Selection

All algorithms we propose are instances of optimistic FTRL with the log-barrier regularizer. The first use of the log-barrier in OPS can be traced back to the barrier subgradient method proposed by Nesterov [26]. Later, Luo et al. [22] employed a hybrid regularizer, which incorporated the log-barrier, in the development of ADA-BARRONS. This marked the first OPS algorithm with a regret rate polylogarithmic in $T$ and an acceptable per-round time complexity of $O(d^{2.5}T)$. Van Erven et al. [35] conjectured that FTRL with the log-barrier (LB-FTRL) achieves the optimal regret. The conjecture was recently refuted by Zimmert et al. [38], who established a regret lower bound for LB-FTRL. Jézéquel et al. [17] combined the log-barrier and volumetric barrier to develop VB-FTRL, the first algorithm with near-optimal regret and an acceptable per-round time complexity of $O(d^2T)$. These regret bounds are worst-case and do not directly imply our results.

### 2.2   FTRL with Self-Concordant Regularizer

Abernethy et al. [1] showed that when the regularizer is chosen as a self-concordant barrier of the constraint set, the regret of FTRL is bounded by the sum of dual local norms of the gradients. Rakhlin and Sridharan [31] generalized this result for optimistic FTRL.

The requirement for the regularizer to be a barrier is restrictive. For instance, while the log-barrier is self-concordant, it is not a barrier of the probability simplex. To address this issue, van Erven et al. [35], Mhammedi and Rakhlin [24], and Jézéquel et al. [17] introduced an affine transformation such that, after the transformation, the log-barrier becomes a self-concordant barrier of the constraint set. Nonetheless, this reparametrization complicates the proofs.

Theorem 3.2 in this paper shows that optimistic FTRL with a self-concordant regularizer, *without the barrier requirement*, still satisfies a regret bound similar to that by Rakhlin and Sridharan [31]. The proof of Theorem 3.2 aligns with the analyses by Mohri and Yang [25], McMahan [23], and Joulani et al. [19] of FTRL with optimism and adaptivity, as well as the local-norm based analysis by Zimmert et al. [38, Appendix H].

In comparison, Theorem 3.2 generalizes the analysis of Zimmert et al. [38, Appendix H] for optimistic algorithms and time-varying learning rates; Theorem 3.2 differs from the analyses of Mohri and Yang

[25], McMahan [23], and Joulani et al. [19] in that they require the regularizer to be strongly convex, whereas the log-barrier is not.

## 2.3 Data-Dependent Bounds

The following is a summary of relevant literature on the three types of data-dependent bounds. For a more comprehensive review, readers may refer to, e.g., the lecture notes of Orabona [28].

- Small-loss bounds, also known as $L^\star$ bounds, were first derived by Cesa-Bianchi et al. [6] for online gradient descent for quadratic losses. Exploiting the self-bounding property, Srebro et al. [33] proved a small-loss bound for convex smooth losses. Orabona et al. [29] proved a logarithmic small-loss bound when the loss functions are not only smooth but also Lipschitz and exp-concave.
- Chiang et al. [8] derived the first gradual-variation bound, bounding the regret by the variation of the gradients over the rounds. Rakhlin and Sridharan [31, 32] interpreted the algorithm proposed by Chiang et al. [8] as optimistic online mirror descent and also proposed optimistic FTRL with self-concordant barrier regularizers. Joulani et al. [18] established a gradual-variation bound for optimistic FTRL.
- Cesa-Bianchi et al. [7] initiated the study of second-order regret bounds. Hazan and Kale [13] derived a regret bound characterized by the empirical variance of loss vectors for online linear optimization. In the presence of the no-junk-bonds assumption, Hazan and Kale [14] proved a regret bound for OPS characterized by the empirical variance of price relatives.

Except for those for specific loss functions, these data-dependent bounds assume either smoothness or Lipschitzness of the loss functions. Nevertheless, both assumptions are violated in OPS.

Recently, Hu et al. [16] established small-loss and gradual-variation bounds in the context of Riemannian online convex optimization. We are unaware of any Riemannian structure on the probability simplex that would render the loss functions in OPS geodesically convex and geodesically smooth. For instance, Appendix B shows that the loss functions in OPS are not geodesically convex on the Hessian manifold induced by the log-barrier.

## 3  Analysis of Optimistic FTRL with Self-Concordant Regularizers

This section presents Theorem 3.2, a general regret bound for optimistic FTRL with regularizers that are self-concordant *but not necessarily barriers*. This regret bound forms the basis for the analyses in the remainder of the paper and, as detailed in Section 2.2, generalizes the results of Rakhlin and Sridharan [31] and Zimmert et al. [38, Appendix H].

Consider the following online linear optimization problem involving two players, LEARNER and REALITY. Let $\mathcal{X} \subseteq \mathbb{R}^d$ be a closed convex set. At the $t$-th round,

- first, LEARNER announces $x_t \in \mathcal{X}$;
- then, REALITY announces a vector $v_t \in \mathbb{R}^d$;
- finally, LEARNER suffers a loss given by $\langle v_t, x_t \rangle$.

For any given time horizon $T \in \mathbb{N}$, the regret $R_T(x)$ is defined as the difference between the cumulative loss of LEARNER and that yielded by the action $x \in \mathcal{X}$; that is,

$$R_T(x) := \sum_{t=1}^{T} \langle v_t, x_t \rangle - \sum_{t=1}^{T} \langle v_t, x \rangle, \quad \forall x \in \mathcal{X}.$$

The objective of LEARNER is to achieve a small regret against all $x \in \mathcal{X}$. Algorithm 1 provides a strategy for LEARNER, called optimistic FTRL.

We focus on the case where the regularizer $\varphi$ is a self-concordant function.

**Definition 3.1** (Self-concordant functions). *A closed convex function $\varphi : \mathbb{R}^d \to (-\infty, \infty]$ with an open domain $\mathrm{dom}\,\varphi$ is said to be $M$-self-concordant if it is three-times continuously differentiable on $\mathrm{dom}\,\varphi$ and*

$$\left| D^3 \varphi(x)[u, u, u] \right| \leq 2M \langle u, \nabla^2 \varphi(x) u \rangle^{3/2}, \quad \forall x \in \mathrm{dom}\,\varphi, u \in \mathbb{R}^d.$$

---
**Algorithm 1** Optimistic FTRL for Online Linear Optimization
---
**Input:** A sequence of learning rates $\{\eta_t\} \subset \mathbb{R}_{++}$.
1: $\hat{v}_1 \leftarrow 0$.
2: $x_1 \in \arg\min_{x \in \mathcal{X}} \eta_0^{-1} \varphi(x)$.
3: **for all** $t \in \mathbb{N}$ **do**
4:     Announce $x_t$ and receive $v_t \in \mathbb{R}^d$.
5:     Choose an estimate $\hat{v}_{t+1}$ for $v_{t+1}$.
6:     $x_{t+1} \leftarrow \arg\min_{x \in \mathcal{X}} \langle v_{1:t}, x \rangle + \langle \hat{v}_{t+1}, x \rangle + \eta_t^{-1} \varphi(x)$.
7: **end for**
---

Suppose that $\nabla^2 \varphi$ is positive definite at a point $x$. The associated local and dual local norms are given by $\|v\|_x := \sqrt{\langle v, \nabla^2\varphi(x)v \rangle}$ and $\|v\|_{x,*} := \sqrt{\langle v, \nabla^{-2}\varphi(x)v \rangle}$, respectively. Define $\omega(t) := t - \log(1+t)$.

**Theorem 3.2.** *Let $\varphi$ be an $M$-self-concordant function such that $\mathcal{X}$ is contained in the closure of $\operatorname{dom}\varphi$ and $\min_{x \in \mathcal{X}} \varphi(x) = 0$. Suppose that $\nabla^2\varphi(x)$ is positive definite for all $x \in \mathcal{X} \cap \operatorname{dom}\varphi$ and the sequence of learning rates $\{\eta_t\}$ is non-increasing. Then, Algorithm 1 satisfies*

$$R_T(x) \le \frac{\varphi(x)}{\eta_T} + \sum_{t=1}^{T} \left( \langle v_t - \hat{v}_t, x_t - x_{t+1} \rangle - \frac{1}{\eta_{t-1}M^2}\omega(M\|x_t - x_{t+1}\|_{x_t}) \right).$$

*If in addition, $\eta_{t-1}\|v_t - \hat{v}_t\|_{x_t,*} \le 1/(2M)$ for all $t \in \mathbb{N}$, then Algorithm 1 satisfies*

$$R_T(x) \le \frac{\varphi(x)}{\eta_T} + \sum_{t=1}^{T} \eta_{t-1}\|v_t - \hat{v}_t\|_{x_t,*}^2.$$

The proof of Theorem 3.2 is deferred to Appendix D. It is worth noting that the crux of the proof lies in Lemma D.1; the remaining steps follow standard procedure.

# 4   "Smoothness" in Online Portfolio Selection

## 4.1   Online Portfolio Selection

Online Portfolio Selection (OPS) is a multi-round game between two players, say INVESTOR and MARKET. Suppose there are $d$ investment alternatives. A portfolio of INVESTOR is represented by a vector in the probability simplex in $\mathbb{R}^d$, which indicates the distribution of INVESTOR's wealth among the $d$ investment alternatives. The price relatives of the investment alternatives at the $t$-th round are listed in a vector $a_t \in \mathbb{R}_+^d$.

The game has $T$ rounds. At the $t$-th round,

- first, INVESTOR announces a portfolio $x_t \in \Delta \subset \mathbb{R}^d$;
- then, MARKET announces the price relatives $a_t \in \mathbb{R}_+^d$;
- finally, INVESTOR suffers a loss given by $f_t(x_t)$, where the loss function $f_t$ is defined as

$$f_t(x) := -\log\langle a_t, x \rangle.$$

The objective of INVESTOR is to achieve a small regret against all portfolios $x \in \Delta$, defined as[2]

$$R_T(x) := \sum_{t=1}^{T} f_t(x_t) - \sum_{t=1}^{T} f_t(x), \quad \forall x \in \Delta \cap \bigcap_{t=1}^{T} \operatorname{dom} f_t.$$

In the context of OPS, the regret corresponds to the logarithm of the ratio between the wealth growth rate of INVESTOR and that yielded by the constant rebalanced portfolio represented by $x \in \Delta$.

**Assumption 1.** *The vector of price relatives $a_t$ is non-zero and satisfies $\|a_t\|_\infty = 1$ for all $t \in \mathbb{N}$.*

---
[2]Because the vectors $a_t$ can have zero entries, in general, $\operatorname{dom} f_t$ does not contain $\Delta$.

The assumption on $\|a_t\|_\infty$ does not restrict the problem's applicability. If the assumption does not hold, then we can consider another OPS game with $a_t$ replaced by $\tilde{a}_t := a_t/\|a_t\|_\infty$ and develop algorithms and define the regret with respect to $\tilde{a}_t$. It is obvious that the regret values defined with $a_t$ and $\tilde{a}_t$ are the same.

The following observation, readily verified by direct calculation, will be useful in the proofs.

**Lemma 4.1.** *The vector $x \odot (-\nabla f_t(x))$ lies in $\Delta$ for all $x \in \mathrm{ri}\,\Delta$ and $t \in \mathbb{N}$.*

### 4.2 Log-Barrier

Standard online convex optimization algorithms, such as those in the lecture notes by Orabona [28] and Hazan [12], assume that the loss functions are either Lipschitz or smooth.

**Definition 4.2** (Lipschitzness and smoothness). *A function $\varphi$ is said to be Lipschitz with respect to a norm $\|\cdot\|$ if*

$$|\varphi(y) - \varphi(x)| \le L\|y - x\|, \quad \forall x, y \in \mathrm{dom}\,\varphi$$

*for some $L > 0$. It is said to be smooth with respect to the norm $\|\cdot\|$ if*

$$\|\nabla\varphi(y) - \nabla\varphi(x)\|_* \le L'\|y - x\|, \quad \forall x, y \in \mathrm{dom}\,\nabla\varphi \tag{1}$$

*for some $L' > 0$, where $\|\cdot\|_*$ denotes the dual norm.*

Given that $\langle a_t, x \rangle$ can be arbitrarily close to zero on $\Delta \cap \mathrm{dom}\,f_t$ in OPS, it is well known that there does not exist a Lipschitz parameter $L$ nor a smoothness parameter $L'$ for all loss functions $f_t$. Therefore, standard online convex optimization algorithms do not directly apply.

We define the log-barrier as[3]

$$h(x) := -d\log d - \sum_{i=1}^d \log x(i), \quad \forall (x(1), \ldots, x(d)) \in \mathbb{R}^d_{++}. \tag{2}$$

It is easily checked that the local and dual local norms associated with the log-barrier are given by

$$\|u\|_x := \|u \oslash x\|_2, \quad \|u\|_{x,*} = \|u \odot x\|_2. \tag{3}$$

In the remainder of the paper, we will only consider this pair of local and dual local norms.

Note that Lipschitzness implies boundedness of the gradient. The following observation motivates the use of the log-barrier in OPS, showing that the gradients in OPS are bounded with respect to the dual local norms defined by the log-barrier.

**Lemma 4.3.** *It holds that $\|\nabla f_t(x)\|_{x,*} \le 1$ for all $x \in \mathrm{ri}\,\Delta$ and $t \in \mathbb{N}$.*

A similar result was proved by van Erven et al. [35, (2)]. We provide a proof of Lemma 4.3 in Section E for completeness.

The following fact will be useful.

**Lemma 4.4** (Nesterov [27, Example 5.3.1 and Theorem 5.3.2]). *The log-barrier $h$ and loss functions $f_t$ in OPS are both $1$-self-concordant.*

### 4.3 "Smoothness" in OPS

Existing results on small-loss and gradual-variation bounds require the loss functions to be smooth. For example, Chiang et al. [8] exploited the definition of smoothness (1) to derive gradual-variation bounds; Srebro et al. [33] and Orabona et al. [29] used the "self-bounding property," a consequence of smoothness, to derive small-loss bounds.

**Lemma 4.5** (Self-bounding property [33, Lemma 2.1]). *Let $f : \mathbb{R}^d \to \mathbb{R}$ be an $L$-smooth convex function with $\mathrm{dom}\,f = \mathbb{R}^d$. Then, $\|\nabla f(x)\|_*^2 \le 2L(f(x) - \min_{y \in \mathbb{R}^d} f(y))$ for all $x \in \mathbb{R}^d$.*

While the loss functions in OPS are not smooth, we provide two smoothness characterizations of the loss functions in OPS. The first is analogous to the definition of smoothness (1).

---

[3]The definition here slightly differs from those typically seen in the literature with an additional $-d\log d$ term. The additional term helps us remove a $\log d$ term in the regret bounds.

**Lemma 4.6.** *Let $f(x) = -\log \langle a, x \rangle$ for some $a \in \mathbb{R}_+^d$. Under Assumption 1 on the vector $a$,*

$$\|(x \odot \nabla f(x)) - (y \odot \nabla f(y))\|_2 \le 4 \min \left\{ \|x - y\|_x, \|x - y\|_y \right\}, \quad \forall x, y \in \mathrm{ri}\, \Delta,$$

*where $\odot$ denotes the entrywise product and $\|\cdot\|_x$ and $\|\cdot\|_y$ are the local-norms defined by the log-barrier* (3).

The second smoothness characterization is analogous to the self-bounding property (Lemma 4.5). For any $x \in \mathrm{ri}\, \Delta$ and $v \in \mathbb{R}^d$, define

$$\alpha_x(v) := \frac{-\sum_{i=1}^d x^2(i) v(i)}{\sum_{i=1}^d x^2(i)}, \quad \forall x \in \mathrm{ri}\, \Delta, \tag{4}$$

where $x(i)$ and $v(i)$ denote the $i$-th entries of $x$ and $v$, respectively.

**Lemma 4.7.** *Let $f(x) = -\log \langle a, x \rangle$ for some $a \in \mathbb{R}_+^d$. Then, under Assumption 1 on the vector $a$, it holds that*

$$\|\nabla f(x) + \alpha_x(\nabla f(x))e\|_{x,*}^2 \le 4f(x), \quad \forall x \in \mathrm{ri}\, \Delta,$$

*where the notation $e$ denotes the all-ones vector and $\|\cdot\|_{x,*}$ denotes the dual local norm defined by the log-barrier* (3).

**Remark 4.8.** *The value $\alpha_x(v)$ is indeed chosen to minimize $\|v + \alpha e\|_{x,*}^2$ over all $\alpha \in \mathbb{R}$.*

The proofs of Lemma 4.6 and Lemma 4.7 are deferred to Appendix E.

## 5 LB-FTRL with Multiplicative-Gradient Optimism

Define $g_t := \nabla f_t(x_t)$. By the convexity of the loss functions, OPS can be reduced to an online linear optimization problem described in Section 3 with $v_t = g_t$ and $\mathcal{X}$ being the probability simplex $\Delta$. Set the regularizer $\varphi$ as the log-barrier (2) in Optimistic FTRL (Algorithm 1). By Lemma 4.4, for the regret guarantee in Theorem 3.2 to be valid, it remains to ensure that $\hat{g}_t$, the estimate of $g_t$, is selected to satisfy $\eta_{t-1}\|g_t - \hat{g}_t\|_{x_t,*} \le 1/2$ for all $t$. However, as $x_t$, which defines the dual local norm, depends on $\hat{g}_t$ in Algorithm 1, selecting such $\hat{g}_t$ is non-trivial.

To address this issue, we introduce Algorithm 2. This algorithm simultaneously computes the next iterate $x_{t+1}$ and the gradient estimate $\hat{g}_{t+1}$ by solving a system of nonlinear equations (5). As we estimate $x_{t+1} \odot g_{t+1}$ instead of $g_{t+1}$, we call the algorithm LB-FTRL with Multiplicative-Gradient Optimism. Here, LB indicates that the algorithm adopts the log-barrier as the regularizer.

---

**Algorithm 2** LB-FTRL with Multiplicative-Gradient Optimism for OPS

---

    **Input:** A sequence of learning rates $\{\eta_t\} \subseteq \mathbb{R}_{++}$.
1: $h(x) := -d \log d - \sum_{i=1}^d \log x(i)$.
2: $\hat{g}_1 := 0$.
3: $x_1 \leftarrow \arg\min_{x \in \Delta} \eta_0^{-1} h(x)$.
4: **for all** $t \in \mathbb{N}$ **do**
5:      Announce $x_t$ and receive $a_t$.
6:      $g_t := \nabla f_t(x_t)$.
7:      Choose an estimate $p_{t+1} \in \mathbb{R}^d$ for $x_{t+1} \odot g_{t+1}$.
8:      Compute $x_{t+1}$ and $\hat{g}_{t+1}$ such that

$$\begin{cases} x_{t+1} \odot \hat{g}_{t+1} = p_{t+1}, \\ x_{t+1} \in \arg\min_{x \in \Delta} \langle g_{1:t}, x \rangle + \langle \hat{g}_{t+1}, x \rangle + \eta_t^{-1} h(x). \end{cases} \tag{5}$$

9: **end for**

---

By the definitions of the dual local norm (3) and $p_t$ (5), we write

$$\|g_t - \hat{g}_t\|_{x_t,*} = \|x_t \odot g_t - x_t \odot \hat{g}_t\|_2 = \|x_t \odot g_t - p_t\|_2.$$

It suffices to choose $p_t$ such that $\eta_{t-1}\|x_t \odot g_t - p_t\|_2 \le 1/2$. Indeed, Algorithm 3 and Algorithm 4 correspond to choosing $p_t = x_{t-1} \odot g_{t-1}$ and $p_t = 0$, respectively. Algorithm 5 in Appendix I corresponds to choosing $p_t = (1/\eta_{0:t-2}) \sum_{\tau=1}^{t-1} \eta_{\tau-1} x_\tau \odot g_\tau$.

Theorem 5.1 guarantees that $x_t$ and $\hat{g}_t$ are well-defined and can be efficiently computed. Its proof and the computational details can be found in Appendix F.1.

**Theorem 5.1.** *If $\eta_t p_{t+1} \in [-1, 0]^d$, then the system of nonlinear equations* (5) *has a solution. The solution can be computed in $\tilde{O}(d)$ time.*

Algorithm 2 corresponds to Algorithm 1 with $v_t = g_t$, $\hat{v}_t = \hat{g}_t = p_t \oslash x_t$, and $\varphi(x) = h(x)$. Corollary 5.2 then follows from Theorem 3.2. Its proof can be found in Appendix F.2.

**Corollary 5.2.** *Assume that the sequence $\{\eta_t\}$ is non-increasing and $p_t \in (-\infty, 0]^d$ for all $t \in \mathbb{N}$. Under Assumption 1, Algorithm 2 satisfies*

$$R_T(x) \le \frac{d \log T}{\eta_T} + \sum_{t=1}^{T} \left( \langle g_t - p_t \oslash x_t, x_t - x_{t+1} \rangle - \frac{1}{\eta_{t-1}} \omega(\|x_t - x_{t+1}\|_{x_t}) \right) + 2,$$

*In addition, for any sequence of vectors $\{u_t\}$ such that $\eta_{t-1}\|(g_t + u_t) \odot x_t - p_t\|_2 \le 1/2$ and $\langle u_t, x_t - x_{t+1} \rangle = 0$ for all $t \in \mathbb{N}$, Algorithm 2 satisfies*

$$R_T(x) \le \frac{d \log T}{\eta_T} + \sum_{t=1}^{T} \eta_{t-1}\|(g_t + u_t) \odot x_t - p_t\|_2^2 + 2.$$

**Remark 5.3.** *The vectors $u_t$ are deliberately introduced to derive a small-loss bound for OPS.*

# 6 Data-Dependent Bounds for OPS

## 6.1 Gradual-Variation Bound

We define the *gradual variation* as

$$V_T := \sum_{t=2}^{T} \|\nabla f_t(x_{t-1}) - \nabla f_{t-1}(x_{t-1})\|_{x_{t-1},*}^2 \le \sum_{t=2}^{T} \max_{x \in \Delta} \|\nabla f_t(x) - \nabla f_{t-1}(x)\|_{x,*}^2, \quad (6)$$

where $\|\cdot\|_*$ denotes the dual local norm associated with the log-barrier. The definition is a local-norm analog to the existing one [8, 18], defined as $\sum_{t=2}^{T} \max_{x \in \Delta} \|\nabla f_t(x) - \nabla f_{t-1}(x)\|^2$ for a *fixed* norm $\|\cdot\|$. Regarding Lemma 4.3, our definition appears to be a natural extension.

In this sub-section, we introduce Algorithm 3, LB-FTRL with Last-Multiplicative-Gradient Optimism, and Theorem 6.1, which provides the first gradual-variation bound for OPS. Algorithm 3 is an instance of Algorithm 2 with $p_1 = 0$ and $p_t = x_{t-1} \odot g_{t-1}$ for $t \ge 2$. Note that the learning rates specified in Theorem 6.1 do not require the knowledge of $V_T$ in advance and can be computed on the fly.

The proof of Theorem 6.1 can be found in Appendix G.

**Theorem 6.1.** *Let $\eta_0 = \eta_1 = 1/(16\sqrt{2})$ and $\eta_t = \sqrt{d/(512d + 2 + V_t)}$ for $t \ge 2$. Then, Algorithm 3 satisfies*

$$R_T(x) \le (\log T + 8)\sqrt{dV_T + 512d^2} + \sqrt{2d}\log T + 2 - 128\sqrt{2d}, \quad \forall T \in \mathbb{N}.$$

By the definition of the dual local norm (3) and Lemma 4.1,

$$V_T = \sum_{t=2}^{T} \|x_{t-1} \odot \nabla f_t(x_{t-1}) - x_{t-1} \odot \nabla f_{t-1}(x_{t-1})\|_2^2 \le 2(T - 1).$$

As a result, the worst-case regret of Algorithm 3 is $O(\sqrt{dT}\log T)$, comparable to the regret bounds of the barrier subgradient method [26], Soft-Bayes [30], and LB-OMD [34] up to logarithmic factors. On the other hand, if the price relatives remain constant over rounds, then $V_T = 0$ and $R_T = O(d \log T)$.

---

**Algorithm 3** LB-FTRL with Last-Multiplicative-Gradient Optimism for OPS

---

    **Input:** A sequence of learning rates $\{\eta_t\} \subseteq \mathbb{R}_{++}$.

1: $h(x) := -d \log d - \sum_{i=1}^{d} \log x(i)$.
2: $p_1 \leftarrow 0$.
3: $x_1 \leftarrow \arg\min_{x \in \mathcal{X}} \eta_0^{-1} h(x)$.
4: **for all** $t \in \mathbb{N}$ **do**
5:     Announce $x_t$ and receive $a_t$.
6:     $g_t \leftarrow \nabla f_t(x_t)$.
7:     $p_{t+1} \leftarrow x_t \odot g_t$.
8:     Compute $x_{t+1}$ and $\hat{g}_{t+1}$ such that

$$\begin{cases} x_{t+1} \odot \hat{g}_{t+1} = p_{t+1}, \\ x_{t+1} \in \arg\min_{x \in \Delta} \langle g_{1:t}, x \rangle + \langle \hat{g}_{t+1}, x \rangle + \eta_t^{-1} h(x). \end{cases}$$

9: **end for**

---

**Time Complexity.** The vectors $g_t$ and $p_{t+1}$, as well as the the quantity $V_t$, can be computed using $O(d)$ arithmetic operations. By Theorem 5.1, the iterate $x_{t+1}$ can be computed in $\tilde{O}(d)$ arithmetic operations. Therefore, the per-round time of Algorithm 3 is $\tilde{O}(d)$.

### 6.2 Small-Loss Bound

In this sub-section, we introduce Algorithm 4, Adaptive LB-FTRL, and Theorem 6.2, the first small-loss bound for OPS. The algorithm is an instance of Algorithm 2 with $p_t = 0$. Then, $\hat{g}_{t+1} = 0$ and $x_{t+1}$ is directly given by Line 8 of Algorithm 4. Note that Theorem 5.1 still applies.

---

**Algorithm 4** Adaptive LB-FTRL for OPS

---

1: $h(x) := -d \log d - \sum_{i=1}^{d} \log x(i)$.
2: $x_1 \leftarrow \arg\min_{x \in \Delta} \eta_0^{-1} h(x)$.
3: **for all** $t \in \mathbb{N}$ **do**
4:     Announce $x_t$ and receive $a_t$.
5:     $g_t \leftarrow \nabla f_t(x_t) = -\frac{a_t}{\langle a_t, x_t \rangle}$.
6:     $\alpha_t \leftarrow \alpha_{x_t}(g_t)$ (see the definition (4)).
7:     $\eta_t \leftarrow \dfrac{\sqrt{d}}{\sqrt{4d+1+\sum_{\tau=1}^{t} \|g_\tau + \alpha_\tau e\|_{x_\tau,*}^2}}$.
8:     $x_{t+1} \leftarrow \arg\min_{x \in \Delta} \langle g_{1:t}, x \rangle + \eta_t^{-1} h(x)$.
9: **end for**

---

The proof of Theorem 6.2 is provided in Appendix H.

**Theorem 6.2.** *Let $L_T^\star = \min_{x \in \Delta} \sum_{t=1}^{T} f_t(x)$. Then, under Assumption 1, Algorithm 4 satisfies*

$$R_T(x) \leq 2(\log T + 2)\sqrt{4dL_T^\star + 4d^2 + d} + d(\log T + 2)^2.$$

Under Assumption 1,

$$L_T^\star = \min_{x \in \Delta} \sum_{t=1}^{T} -\log \langle a_t, x \rangle \leq \sum_{t=1}^{T} -\log \frac{\|a\|_1}{d} = \sum_{t=1}^{T} \log d = T \log d.$$

Assuming $T > d$, the worst-case regret bound is $O(\sqrt{dT \log d} \log T)$, also comparable to the regret bounds of the barrier subgradient method [26], Soft-Bayes [30], and LB-OMD [34] up to logarithmic factors. On the other hand, suppose that there exists an $i^\star \in [d]$ such that $a_t(i^\star) = 1$ for all $t \in [T]$. That is, the $i^\star$-th investment alternative always outperforms all the other investment alternatives. Then, $L_T^\star = 0$ and $R_T = O(d \log^2 T)$.

Assumption 1 does not restrict the applicability of Theorem 6.2, as mentioned earlier. If the assumption does not hold, Theorem 6.2 is applied with respect to the normalized price relatives $\tilde{a}_t = a_t/\|a_t\|_\infty$. In this case, $L_T^\star$ is defined with respect to $\{\tilde{a}_t\}$.

**Time Complexity.** Since $\|g_t + \alpha_t e\|_{x_t,*}^2 = \|x_t \odot g_t + \alpha_t x_t\|_2^2$, it is obvious that computing $\alpha_t$, $\eta_t$, and $g_{1:t}$ can be done in $O(d)$ arithmetic operations. By Theorem 5.1, the iterate $x_{t+1}$ can be computed in $\tilde{O}(d)$ arithmetic operations. Hence, the per-round time of Algorithm 4 is $\tilde{O}(d)$.

## 7 Concluding Remarks

We have presented Theorem 6.1 and Theorem 6.2, the first gradual-variation and small-loss bounds for OPS that do not require the no-junk-bonds assumption, respectively. The algorithms exhibit sublinear regrets in the worst cases and achieve logarithmic regrets in the best cases, with per-round time almost linear in the dimension. They mark the first data-dependent bounds for non-Lipschitz non-smooth losses.

A potential direction for future research is to extend our analyses for a broader class of online convex optimization problems. In particular, it remains unclear how to extend the two smoothness characterizations (Lemma 4.6 and Lemma 4.7) for other loss functions.

Orabona et al. [29] showed that achieving a regret rate of $O(d^2 + \log L_T^\star)$ is possible under the no-junk-bonds assumption, where $L_T^\star$ denotes the cumulative loss of the best constant rebalanced portfolio. This naturally raises the question: can a similar regret rate be attained without relying on the no-junk-bonds assumption? If so, then the regret rate will be constant in $T$ in the best cases and logarithmic in $T$ in the worst cases. However, considering existing results in probability forecasting with the logarithmic loss [5, Chapter 9]—a special case of OPS without the no-junk-bonds assumption—such a data-dependent regret rate seems improbable. Notably, classical rate-optimal algorithms for probability forecasting with the logarithmic loss, such as Shtarkov's minimax-optimal algorithm, the Laplace mixture, and the Krichevsky-Trofimov mixture, all achieve logarithmic regret rates for all possible data sequences [5, Chapter 9].

Zhao et al. [37] showed that for Lipschitz and smooth losses, an algorithm with a gradual-variation bound automatically achieves a small-loss bound. Generalizing their argument for non-Lipschitz non-smooth losses is a natural direction to consider.

## Acknowledgments and Disclosure of Funding

The authors are supported by the Young Scholar Fellowship (Einstein Program) of the National Science and Technology Council of Taiwan under grant number NSTC 112-2636-E-002-003, by the 2030 Cross-Generation Young Scholars Program (Excellent Young Scholars) of the National Science and Technology Council of Taiwan under grant number NSTC 112-2628-E-002-019-MY3, and by the research project "Pioneering Research in Forefront Quantum Computing, Learning and Engineering" of National Taiwan University under grant number NTU-CC-112L893406.

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

## A  A Summary of OPS Algorithms

Table 1: A summary of existing algorithms for online portfolio selection without the no-junk-bonds assumption. Assume $T \gg d$.

| Algorithms | Regret ($\tilde{O}$) | | Per-round time ($\tilde{O}$) |
|---|---|---|---|
| | Best-case | Worst-case | |
| Universal Portfolio [9, 20] | $d \log T$ | | $d^4 T^{14}$ |
| $\widetilde{\text{EG}}$ [15, 34] | $d^{1/3} T^{2/3}$ | | $d$ |
| BSM [26], Soft-Bayes [30], LB-OMD [34] | $\sqrt{dT}$ | | $d$ |
| ADA-BARRONS [22] | $d^2 \log^4 T$ | | $d^{2.5} T$ |
| LB-FTRL without linearized losses [35] | $d \log^{d+1} T$ | | $d^2 T$ |
| PAE+DONS [24] | $d^2 \log^5 T$ | | $d^3$ |
| BISONS [38] | $d^2 \log^2 T$ | | $d^3$ |
| VB-FTRL [17] | $d \log T$ | | $d^2 T$ |
| **This work** (Algorithm 3) | $d \log T$ | $\sqrt{dT}$ | $d$ |
| **This work** (Algorithm 4) | $d \log^2 T$ | $\sqrt{dT}$ | $d$ |

## B  Loss Functions in OPS are not Geodesically Convex

Let $h(x) := -d \log d - \sum_{i=1}^d \log x(i)$ be the log-barrier. Let $(\mathcal{M}, g)$ be the Hessian manifold induced by the log-barrier, where $\mathcal{M} = \operatorname{ri} \Delta$ and $g$ is the Riemannian metric defined by $\langle u, v \rangle_x := \langle u, \nabla^2 h(x) v \rangle$.

**Lemma B.1** (Gzyl and Nielsen [11, Theorem 3.1]). *Let $x, y \in \operatorname{ri} \Delta$. The geodesic $\gamma : [0,1] \to \operatorname{ri} \Delta$ connecting $x, y$ is given by*

$$\gamma(t)(i) = \frac{x(i)^{1-t} y(i)^t}{\sum_{j=1}^d x(j)^{1-t} y(j)^t}, \quad \forall i \in [d],$$

*where $\gamma(t)(i)$ denotes the $i$-th entry of the vector $\gamma(t) \in \operatorname{ri} \Delta$.*

The proposition below shows that the loss functions in OPS are not geodesically convex on $(\mathcal{M}, g)$ in general.

**Proposition B.2.** *Let $a = (2, 1) \in \mathbb{R}_{++}^2$. The function $f(x) = -\log \langle a, x \rangle$ is not geodesically convex on $(\mathcal{M}, g)$.*

*Proof.* It suffices to show that $f \circ \gamma : [0, 1] \to \mathbb{R}$ is not a convex function [4, Definition 11.3]. Let $x = (1/2, 1/2)$ and $y = (1/(1 + e), e/(1 + e))$. By Lemma B.1, the geodesic connecting $x, y$ is $\gamma(t) = (1/(1 + e^t), e^t/(1 + e^t))$. We write

$$(f \circ \gamma)''(t) = \frac{(2 - e^{2t}) e^t}{(2 + e^t)^2 (1 + e^t)^2} < 0, \quad \forall t > \frac{\log 2}{2},$$

showing that $f \circ \gamma$ is not convex. The proposition follows. $\qquad \square$

## C  Properties of Self-Concordant Functions

This section reviews properties of self-concordant functions relevant to our proofs. Readers are referred to the book by Nesterov [27] for a complete treatment.

Throughout this section, let $\varphi$ be a self-concordant function (Definition 3.1) and $\|\cdot\|_x$ be the associated local norm, i.e.,

$$\|u\|_x := \sqrt{\langle u, \nabla^2 \varphi(x) u \rangle}, \quad \forall u \in \mathbb{R}^d, x \in \mathrm{dom}\,\varphi.$$

Self-concordant functions are neither smooth nor strongly convex in general. The following theorem indicates that nevertheless, they are locally smooth and strongly convex.

**Theorem C.1** (Nesterov [27, Theorem 5.1.7]). *For any $x, y \in \mathrm{dom}\,\varphi$ such that $r := \|y - x\|_x < 1/M$,*

$$(1 - Mr)^2 \nabla^2 \varphi(x) \le \nabla^2 \varphi(y) \le \frac{1}{(1 - Mr)^2} \nabla^2 \varphi(x).$$

Define $\omega(t) := t - \log(1 + t)$. Let $\omega_*$ be its Fenchel conjugate, i.e., $\omega_*(t) = -t - \log(1 - t)$.

**Theorem C.2** (Nesterov [27, Theorem 5.1.8 and 5.1.9]). *For any $x, y \in \mathrm{dom}\,\varphi$ with $r := \|y - x\|_x$,*

$$\langle \nabla \varphi(y) - \nabla \varphi(x), y - x \rangle \ge \frac{r^2}{1 + Mr},$$

$$\varphi(y) \ge \varphi(x) + \langle \nabla \varphi(x), y - x \rangle + \frac{1}{M^2}\omega(Mr). \tag{7}$$

*If $r < 1/M$, then*

$$\langle \nabla \varphi(y) - \nabla \varphi(x), y - x \rangle \le \frac{r^2}{1 - Mr},$$

$$\varphi(y) \le \varphi(x) + \langle \nabla \varphi(x), y - x \rangle + \frac{1}{M^2}\omega_*(Mr). \tag{8}$$

The following proposition bounds the growth of $\omega$ and $\omega_*$.

**Proposition C.3** (Nesterov [27, Lemma 5.1.5]). *For any $t \ge 0$,*

$$\frac{t^2}{2(1 + t)} \le \omega(t) \le \frac{t^2}{2 + t}.$$

*For any $t \in [0, 1)$,*

$$\frac{t^2}{2 - t} \le \omega_*(t) \le \frac{t^2}{2(1 - t)}.$$

It is easily checked that both the loss functions in OPS and the log-barrier are not only self-concordant but also self-concordant barriers.

**Definition C.4.** *A $1$-self-concordant function $\varphi$ is said to be a $\nu$-self-concordant barrier if*

$$\langle \nabla \varphi(x), u \rangle^2 \le \nu \langle u, \nabla^2 \varphi(x) u \rangle, \quad \forall x \in \mathrm{dom}\,\varphi, u \in \mathbb{R}^d.$$

**Lemma C.5** (Nesterov [27, (5.4.3)]). *The log-barrier (2) is a $d$-self-concordant barrier.*

# D  Proof of Theorem 3.2 (Regret Analysis of Optimistic FTRL With Self-Concordant Regularizers)

## D.1  Proof of Theorem 3.2

The proof of the following stability lemma is deferred to the next subsection.

**Lemma D.1.** *Define $F_{t+1}(x) := \langle v_{1:t}, x \rangle + \eta_t^{-1} \varphi(x)$. Let $\{\eta_t\}$ be a non-increasing sequence of strictly positive numbers such that $\eta_{t-1}\|v_t - \hat{v}_t\|_{x_t, *} \le 1/(2M)$. Then, Algorithm 1 satisfies*

$$F_t(x_t) - F_{t+1}(x_{t+1}) + \langle v_t, x_t \rangle \le \langle v_t - \hat{v}_t, x_t - x_{t+1} \rangle - \frac{1}{\eta_{t-1}M^2}\omega(M\|x_{t+1} - x_t\|_{x_t})$$

$$\le \frac{1}{\eta_{t-1}M^2}\omega_*(\eta_{t-1}M\|v_t - \hat{v}_t\|_{x_t, *})$$

$$\le \eta_{t-1}\|v_t - \hat{v}_t\|_{x_t, *}^2.$$

Define $\hat{F}_t(x) := F_t(x) + \langle \hat{v}_t, x \rangle$. Since $\hat{v}_{T+1}$ has no effect on the regret, we set $\hat{v}_{T+1} = 0$. By the strong FTRL lemma [28, Lemma 7.1],

$$R_T(x) = \sum_{t=1}^{T} (\langle v_t, x_t \rangle - \langle v_t, x \rangle)$$

$$= \langle \hat{v}_{T+1}, x \rangle + \frac{\varphi(x)}{\eta_T} - \min_{x \in \mathcal{X}} \frac{\varphi(x)}{\eta_0} + \sum_{t=1}^{T} (\hat{F}_t(x_t) - \hat{F}_{t+1}(x_{t+1}) + \langle v_t, x_t \rangle)$$

$$+ \hat{F}_{T+1}(x_{T+1}) - \hat{F}_{T+1}(x)$$

$$\leq \frac{\varphi(x)}{\eta_T} + \sum_{t=1}^{T} (\hat{F}_t(x_t) - \hat{F}_{t+1}(x_{t+1}) + \langle v_t, x_t \rangle)$$

$$\leq \frac{\varphi(x)}{\eta_T} + \sum_{t=1}^{T} (F_t(x_t) - F_{t+1}(x_{t+1}) + \langle v_t, x_t \rangle).$$

The penultimate line above follows from the assumption that $\min_{x \in \mathcal{X}} \varphi(x) = 0$ and that $x_{T+1}$ minimizes $\hat{F}_{T+1}$ on $\mathcal{X}$; the last line above follows from a telescopic sum and $\hat{v}_1 = \hat{v}_{T+1} = 0$. The theorem then follows from Lemma D.1.

### D.2 Proof of Lemma D.1

Since $\varphi$ is $M$-self-concordant, the function $\eta_{t-1} F_t$ is $M$-self-concordant. By Theorem C.2,

$$\eta_{t-1} F_t(x_{t+1}) - \eta_{t-1} F_t(x_t)$$

$$\geq \langle \eta_{t-1} \nabla F_t(x_t), x_{t+1} - x_t \rangle + \frac{1}{M^2} \omega(M\|x_t - x_{t+1}\|_{x_t})$$

$$= \langle \eta_{t-1}(\nabla F_t(x_t) + \hat{v}_t), x_{t+1} - x_t \rangle - \eta_{t-1} \langle \hat{v}_t, x_{t+1} - x_t \rangle + \frac{1}{M^2} \omega(M\|x_t - x_{t+1}\|_{x_t}).$$

Since $x_t$ minimizes $F_t(x) + \langle \hat{v}_t, x \rangle$ on $\mathcal{X}$, the optimality condition implies

$$\langle \nabla F_t(x_t) + \hat{v}_t, x_{t+1} - x_t \rangle \geq 0.$$

Then, we obtain

$$\eta_{t-1} F_t(x_{t+1}) - \eta_{t-1} F_t(x_t) \geq -\eta_{t-1} \langle \hat{v}_t, x_{t+1} - x_t \rangle + \frac{1}{M^2} \omega(M\|x_{t+1} - x_t\|_{x_t}).$$

Next, by the non-increasing of $\{\eta_t\}$, $\varphi(x_{t+1}) \geq 0$, and the last inequality,

$$F_t(x_t) - F_{t+1}(x_{t+1}) + \langle v_t, x_t \rangle$$

$$= F_t(x_t) - F_t(x_{t+1}) + \langle v_t, x_t \rangle - \langle v_t, x_{t+1} \rangle + \left(\frac{1}{\eta_{t-1}} - \frac{1}{\eta_t}\right) \varphi(x_{t+1})$$

$$\leq F_t(x_t) - F_t(x_{t+1}) + \langle v_t, x_t - x_{t+1} \rangle$$

$$\leq \langle v_t - \hat{v}_t, x_t - x_{t+1} \rangle - \frac{1}{\eta_{t-1} M^2} \omega(M\|x_{t+1} - x_t\|_{x_t}).$$

This proves the first inequality in the lemma.

By Hölder's inequality,

$$F_t(x_t) - F_{t+1}(x_{t+1}) + \langle v_t, x_t \rangle \leq \|v_t - \hat{v}_t\|_{x_t,*} \|x_t - x_{t+1}\|_{x_t} - \frac{1}{\eta_{t-1} M^2} \omega(M\|x_t - x_{t+1}\|_{x_t}). \tag{9}$$

By the Fenchel-Young inequality,

$$\omega(M\|x_t - x_{t+1}\|_{x_t}) + \omega_*(\eta_{t-1} M\|v_t - \hat{v}_t\|_{x_t,*}) \geq \eta_{t-1} M^2 \|x_{t+1} - x_t\|_{x_t} \|v_t - \hat{v}_t\|_{x_t,*}. \tag{10}$$

Combining (9) and (10) yields

$$F_t(x_t) - F_{t+1}(x_{t+1}) + \langle v_t, x_t \rangle \leq \frac{1}{\eta_{t-1} M^2} \omega_*(\eta_{t-1} M\|v_t - \hat{v}_t\|_{x_t,*}).$$

which proves the second inequality. The third inequality in the lemma then follows from the assumption that $\eta_{t-1}\|v_t - \hat{v}_t\|_{x_t,*} \leq 1/(2M)$ and Proposition C.3.

# E  Proofs in Section 4 (Smoothness Characterizations)

## E.1  Proof of Lemma 4.3

We write

$$\|\nabla f(x)\|_{x,*}^2 = \frac{\sum_{i=1}^d a(i)^2 x(i)^2}{\left(\sum_{i=1}^d a(i)x(i)\right)^2} \le 1.$$

## E.2  Proof of Lemma 4.6

Define $r := \|x - y\|_y$. Consider the following two cases.

(i) Suppose that $r \ge 1/2$. By Lemma 4.1,

$$\|x \odot \nabla f(x) - y \odot \nabla f(y)\|_2 \le \sqrt{2} \le 4 \cdot \frac{1}{2} \le 4\|x - y\|_y.$$

(ii) Suppose that $r < 1/2$. We write

$$r^2 = \|x - y\|_y^2 = \sum_{i=1}^d \left(1 - \frac{x(i)}{y(i)}\right)^2 \ge \left(1 - \frac{x(i)}{y(i)}\right)^2, \quad \forall i \in [d],$$

showing that

$$0 < 1 - r \le \frac{x(i)}{y(i)} \le 1 + r, \quad \forall i \in [d].$$

Then [10, Lemma 1],

$$\frac{\langle a, y \rangle}{\langle a, x \rangle} = \frac{\sum_i a(i)y(i)}{\sum_i a(i)x(i)} \le \max_{i \in [d]} \frac{a(i)y(i)}{a(i)x(i)} = \max_{i \in [d]} \frac{y(i)}{x(i)} \le \frac{1}{1 - r};$$

similarly,

$$\frac{\langle a, y \rangle}{\langle a, x \rangle} \ge \frac{1}{1 + r}.$$

We obtain

$$\frac{-2r}{1 - r} = 1 - \frac{1 + r}{1 - r} \le 1 - \frac{x(i)\langle a, y \rangle}{y(i)\langle a, x \rangle} \le 1 - \frac{1 - r}{1 + r} = \frac{2r}{1 + r}.$$

Since $r < 1/2$,

$$\left(1 - \frac{x(i)\langle a, y \rangle}{y(i)\langle a, x \rangle}\right)^2 \le \max\left\{\left(\frac{-2r}{1 - r}\right)^2, \left(\frac{2r}{1 + r}\right)^2\right\} = \frac{4r^2}{(1 - r)^2} < 16r^2.$$

Therefore,

$$\|x \odot \nabla f(x) - y \odot \nabla f(y)\|_2^2 = \sum_{i=1}^d \left(\frac{a(i)x(i)}{\langle a, x \rangle} - \frac{a(i)y(i)}{\langle a, y \rangle}\right)^2$$

$$= \sum_{i=1}^d \left(\frac{a(i)y(i)}{\langle a, y \rangle}\right)^2 \left(1 - \frac{x(i)\langle a, y \rangle}{y(i)\langle a, x \rangle}\right)^2$$

$$< 16r^2 \sum_{i=1}^d \left(\frac{a(i)y(i)}{\langle a, y \rangle}\right)^2$$

$$< 16r^2,$$

showing that $\|x \odot \nabla f(x) - y \odot \nabla f(y)\|_2 < 4\|x - y\|_y$.

Combining the two cases, we obtain $\|x \odot \nabla f(x) - y \odot \nabla f(y)\|_2 \le 4\|x - y\|_y$. Since $x$ and $y$ are symmetric, we also have $\|x \odot \nabla f(x) - y \odot \nabla f(y)\|_2 \le 4\|x - y\|_x$. This completes the proof.

### E.3 Proof of Lemma 4.7

We will use the notion of relative smoothness.

**Definition E.1** (Bauschke et al. [3], Lu et al. [21])**.** *A function $f$ is said to be $L$-smooth relative to a function $h$ if the function $Lh - f$ is convex.*

**Lemma E.2** (Bauschke et al. [3, Lemma 7])**.** *Let $f(x) = -\log \langle a, x \rangle$ for some non-zero $a \in \mathbb{R}_+^d$. Then, $f$ is $1$-smooth relative to the logarithmic barrier $h$.*

Fix $x \in \mathrm{ri}\,\Delta$. By Lemma E.2, the function $h - f$ is convex and hence

$$h(x) - f(x) + \langle \nabla h(x) - \nabla f(x), v \rangle \le h(x + v) - f(x + v), \quad \forall v \in \mathrm{ri}\,\Delta - x,$$

where $\mathrm{ri}\,\Delta - x := \{u - x | u \in \mathrm{ri}\,\Delta\}$. By Lemma 4.4, Theorem C.2, and Proposition C.3, if $\|v\|_x \le 1/2$,

$$h(x + v) - h(x) - \langle \nabla h(x), v \rangle \le \omega_*(\|v\|_x) \le \|v\|_x^2.$$

Combining the two inequalities above yields

$$\langle -\nabla f(x), v \rangle - \|v\|_x^2 \le f(x) - f(x + v).$$

Since $f(x + v) \ge \min_{y \in \Delta} f(y) = 0$ (Assumption 1) and $\langle e, v \rangle = 0$, we write

$$\langle -\nabla f(x) - \alpha e, v \rangle - \|v\|_x^2 \le f(x), \quad \forall \alpha \in \mathbb{R}, v \in \mathrm{ri}\,\Delta - x \text{ such that } \|v\|_x \le 1/2.$$

The left-hand side is continuous in $v$, so the condition that $v \in \mathrm{ri}\,\Delta - x$ can be relaxed to $v \in \Delta - x := \{u - x | u \in \Delta\}$. We get

$$\langle -\nabla f(x) - \alpha e, v \rangle - \|v\|_x^2 \le f(x), \quad \forall \alpha \in \mathbb{R}, v \in \Delta - x \text{ such that } \|v\|_x \le 1/2. \tag{11}$$

We show that choosing

$$v = -c\nabla^{-2}h(x)(\nabla f(x) + \alpha_x(\nabla f(x))e), \tag{12}$$

for any $c \in (0, 1/2]$ ensures that $v \in \Delta - x$ and $\|v\|_x \le 1/2$.

- First, we check whether $\|v\|_x \le 1/2$. We write

$$\begin{aligned}
\|v\|_x &= c\|\nabla^{-2}h(x)(\nabla f(x) + \alpha_x(\nabla f(x))e)\|_x \\
&= c\|\nabla f(x) + \alpha_x(\nabla f(x))e\|_{x,*} \\
&\le c\|\nabla f(x)\|_{x,*} \\
&\le c \\
&\le 1/2,
\end{aligned}$$

  where the last line follows from Remark 4.8 and Lemma 4.3.

- Then, we check whether $v \in \Delta - x$ or, equivalently, whether $v + x \in \Delta$. By the definition of $\alpha_x(\nabla f(x))$ (4), each entry of $v$ is given by

$$v_i = -cx(i)^2 \left( \nabla_i f(x) - \frac{\sum_j x(j)^2 \nabla_j f(x)}{\sum_j x(j)^2} \right), \quad \forall i \in [d],$$

  where $\nabla_j f(x)$ is the $j$-th entry of $\nabla f(x)$. Obviously,

$$\sum_{i=1}^d v(i) = \sum_{i=1}^d -cx(i)^2 \left( \nabla_i f(x) - \frac{\sum_j x(j)^2 \nabla_j f(x)}{\sum_j x(j)^2} \right) = 0,$$

  so $\sum_i x(i) + v(i) = 1$. It remains to check whether $x(i) + v(i) \ge 0$ for all $i \in [d]$. Notice that $\nabla f(x)$ is entrywise negative. Then,

$$\begin{aligned}
x(i) + v(i) &= x(i) - cx(i)^2 \left( \nabla_i f(x) - \frac{\sum_j x(j)^2 \nabla_j f(x)}{\sum_j x(j)^2} \right) \\
&\ge x(i) + cx(i)^2 \frac{\sum_j x(j)^2 \nabla_j f(x)}{\sum_j x(j)^2}.
\end{aligned}$$

To show that the lower bound is non-negative, we write

$$-x(i)^2 \frac{\sum_j x(j)^2 \nabla_j f(x)}{\sum_j x(j)^2} = \frac{x(i)^2 \sum_j a(j) x(j)^2}{\sum_j x(j)^2 \langle a, x \rangle}$$

$$\leq \frac{x(i)^2}{\sum_j x(j)^2} \max_{j \in [d]} x(j)$$

$$= x(i) \cdot \max_j \frac{x(i) x(j)}{\sum_k x(k)^2}.$$

If $i = j$, then $\max_j \frac{x(i) x(j)}{\sum_k x(k)^2} \leq 1$; otherwise,

$$\max_{j \neq i} \frac{x(i) x(j)}{\sum_k x(k)^2} \leq \max_{j \neq i} \frac{x(i) x(j)}{x(i)^2 + x(j)^2} \leq \frac{1}{2}.$$

Therefore,

$$x(i) + v(i) \geq (1 - c) x(i) \geq \frac{1}{2} x(i) \geq 0, \quad \forall i \in [d].$$

Plug the chosen $v$ (12) into the inequality (11). We write

$$f(x) \geq \sup_{0 \leq c \leq 1/2} c \|\nabla f(x) + \alpha_x(\nabla f(x))e\|_{x,*}^2 - c^2 \|\nabla f(x) + \alpha_x(\nabla f(x))e\|_{x,*}^2$$

$$= \sup_{0 \leq c \leq 1/2} (c - c^2) \|\nabla f(x) + \alpha_x(\nabla f(x))e\|_{x,*}^2$$

$$= \frac{1}{4} \|\nabla f(x) + \alpha_x(\nabla f(x))e\|_{x,*}^2.$$

This completes the proof.

# F   Proof in Section 5 (LB-FTRL with Multiplicative-Gradient Optimism)

## F.1   Proof of Theorem 5.1

We first check existence of a solution. For convenience, we omit the subscripts in $p_{t+1}$ and $\eta_t$ and write $g$ for $g_{1:t}$.

**Proposition F.1.** *Assume that $\eta p \in [-1, 0]^d$. Define*

$$\lambda^\star \in \underset{\lambda \in \text{dom } \psi}{\arg\min} \, \psi(\lambda), \quad \psi(\lambda) := \lambda - \sum_{i=1}^d (1 - \eta p(i)) \log(\lambda + \eta g(i)). \tag{13}$$

*Then, $\lambda^\star$ exists and is unique. The pair $(x_{t+1}, \hat{g}_{t+1})$, defined by*

$$x_{t+1}(i) := \frac{1 - \eta p(i)}{\lambda^\star + \eta g(i)}, \quad \hat{g}_{t+1}(i) := p(i) \cdot \frac{\lambda^\star + \eta g(i)}{1 - \eta p(i)}, \tag{14}$$

*solves the system of nonlinear equations (5).*

We will use the following observation [27, Theorem 5.1.1].

**Lemma F.2.** *The function $\psi$ is $1$-self-concordant.*

*Proof of Proposition F.1.* By Lemma F.2 and the fact that $\text{dom } \psi$ does not contain any line, the minimizer $\lambda^\star$ exists and is unique [27, Theorem 5.1.13]; sine $\eta p \in [-1, 0]^d$ and $\lambda \in \text{dom } \psi$, we have $x_{t+1}(i) \geq 0$; moreover,

$$\psi'(\lambda^\star) = 1 - \sum_{i=1}^d \frac{1 - \eta p(i)}{\lambda^\star + \eta g(i)} = 0.$$

Then,

$$\sum_{i=1}^d x_{t+1}(i) = 1 - \psi'(\lambda^\star) = 1,$$

showing that $x_{t+1} \in \Delta$.

Then, we check whether $(x_{t+1}, \hat{g}_{t+1})$ satisfies the two equalities (5). It is obvious that $x_{t+1} \odot \hat{g}_{t+1} = p$. By the method of Lagrange multiplier, if there exists $\lambda \in \mathbb{R}$ such that

$$\eta g(i) + \eta \hat{g}_{t+1}(i) - \frac{1}{x_{t+1}(i)} + \lambda = 0, \quad \sum_{i=1}^{d} x_{t+1}(i) = 1, \quad x_{t+1}(i) \geq 0,$$

then $x_{t+1} \in \arg\min_{x \in \Delta} \langle g, x \rangle + \langle \hat{g}_{t+1}, x \rangle + \eta^{-1} h(x)$. It is easily checked that the above conditions are satisfied with $\lambda = \lambda^\star$ and (14). The proposition follows. $\qquad\square$

We now analyze the time complexity. Nesterov [26, Section 7] [27, Appendix A.2] proved that when $p = 0$, Newton's method for solving (13) reaches the region of quadratic convergence of the intermediate Newton's method [27, Section 5.2.1] after $O(\log d)$ iterations. A generalization for possibly non-zero $p$ is provided in Proposition F.3 below. The proof essentially follows the approach of Nesterov [27, Appendix A.2]. It is worth noting that while the function $\psi$ is self-concordant, directly applying existing results on self-concordant minimization by Newton's method [27, Section 5.2.1] does not yield the desired $O(\log d)$ iteration complexity bound.

Starting with an initial iterate $\lambda_0 \in \mathbb{R}$, Newton's method for the optimization problem (13) iterates as

$$\lambda_{t+1} = \lambda_t - \psi'(\lambda_t)/\psi''(\lambda_t), \quad \forall t \in \{0\} \cup \mathbb{N}. \tag{15}$$

**Proposition F.3.** *Assume that $\eta p \in [-1, 0]^d$. Define $i^\star \in \arg\min_{i \in [d]}(-g(i))$. Then, Newton's method with the initial iterate*

$$\lambda_0 = 1 - \eta g(i^\star)$$

*enters the region of quadratic convergence of the intermediate Newton's method [27, Section 5.2.1] after $O(\log d)$ iterations.*

Firstly, we will establish Lemma F.4 and Lemma F.5. These provide characterizations of $\psi'$ and $\lambda_t$ that are necessary for the proof of Proposition F.3.

**Lemma F.4.** *The following hold.*

  *(i) $\psi'$ is a strictly increasing and strictly concave function.*

  *(ii) $\psi'(\lambda^\star) = 0$.*

  *(iii) $\lambda_0 \leq \lambda^\star$.*

*Proof of Lemma F.4.* The first item follows by a direct calculation:

$$\psi''(\lambda) = \sum_{i=1}^{d} \frac{1 - \eta p(i)}{(\lambda + \eta g(i))^2} > 0, \quad \psi'''(\lambda) = -\sum_{i=1}^{d} \frac{2(1 - \eta p(i))}{(\lambda + \eta g(i))^3} < 0.$$

The second item has been verified in the proof of Proposition F.1. As for the third item, we write

$$\psi'(\lambda_0) = 1 - \sum_{i=1}^{d} \frac{1 - \eta p(i)}{\lambda_0 + \eta g(i)} \leq 1 - \frac{1 - \eta p(i^\star)}{\lambda_0 + \eta g(i^\star)} = \eta p(i^\star) \leq 0 = \psi'(\lambda^\star).$$

The third item then follows from $\psi'(\lambda_0) \leq \psi'(\lambda^\star)$ and the first item. $\qquad\square$

**Lemma F.5.** *Let $\{\lambda_t\}$ be the sequence of iterates of Newton's method (15). For any $t \geq 0$, $\psi'(\lambda_t) \leq 0$ and $\lambda_t \leq \lambda^\star$.*

*Proof of Lemma F.5.* Recall that $\operatorname{dom}\psi = (-g(i^\star), \infty)$. We proceed by induction. Obviously, $\lambda_0 \in \operatorname{dom}\psi$. By Lemma F.4, the lemma holds for $t = 0$. Assume that the lemma holds for some non-negative integer $t$. By the iteration rule of Newton's method (15) and the assumption that $\psi'(\lambda_t) \leq 0$,

$$\lambda_{t+1} = \lambda_t - \frac{\psi'(\lambda_t)}{\psi''(\lambda_t)} \geq \lambda_t,$$

which ensures that $\lambda_{t+1} \in \text{dom}\,\psi$. By Lemma F.4 (i) and the iteration rule of Newton's method (15), we have

$$\psi'(\lambda_{t+1}) \le \psi'(\lambda_t) + \psi''(\lambda_t)(\lambda_{t+1} - \lambda_t) = \psi'(\lambda_t) - \psi''(\lambda_t)\frac{\psi'(\lambda_t)}{\psi''(\lambda_t)} = 0.$$

By Lemma F.4 (i) and Lemma F.4 (ii), $\lambda_{t+1} \le \lambda^\star$. The lemma follows. $\qquad\square$

*Proof of Proposition F.3.* Define the Newton decrement as

$$\delta(\lambda) := \frac{|\psi'(\lambda_t)|}{\left|\psi'(\lambda)/\sqrt{\psi''(\lambda)}\right|}.$$

By Lemma F.2, the region of quadratic convergence is [27, Theorem 5.2.2]

$$\mathcal{Q} := \{\lambda \in \text{dom}\,\psi : \delta(\lambda) < 1/2\}.$$

Define $T := \lceil (9 + 7\log_2 d)/4 \rceil$. By Lemma F.5, $\psi'(\lambda_t) \le 0$ for all $t \ge 0$. If $\psi'(\lambda_t) = 0$ for some $0 \le t \le T$, then Newton's method reaches $\mathcal{Q}$ within $T$ steps and the proposition follows. Now, let us assume that $\psi'(\lambda_t) < 0$ for all $0 \le t \le T$. By the concavity of $\psi'$ (Lemma F.4 (i)),

$$\psi'(\lambda_{t-1}) \le \psi'(\lambda_t) + \psi''(\lambda_t)(\lambda_{t-1} - \lambda_t) = \psi'(\lambda_t) + \frac{\psi''(\lambda_t)}{\psi''(\lambda_{t-1})}\psi'(\lambda_{t-1}).$$

Divide both sides by $\psi'(\lambda_{t-1})$. By Lemma F.5 and the AM-GM inequality,

$$1 \ge \frac{\psi'(\lambda_t)}{\psi'(\lambda_{t-1})} + \frac{\psi''(\lambda_t)}{\psi''(\lambda_{t-1})} \ge 2\sqrt{\frac{\psi'(\lambda_t)\psi''(\lambda_t)}{\psi'(\lambda_{t-1})\psi''(\lambda_{t-1})}}.$$

Then, for all $1 \le t \le T+1$,

$$|\psi'(\lambda_t)\psi''(\lambda_t)| \le \frac{1}{4}|\psi'(\lambda_{t-1})\psi''(\lambda_{t-1})|.$$

This implies that for all $0 \le t \le T+1$,

$$|\psi'(\lambda_t)\psi''(\lambda_t)| \le \frac{1}{4^t}|\psi'(\lambda_0)\psi''(\lambda_0)|,$$

which further implies that

$$\delta(\lambda_t)^2 = \frac{(\psi'(\lambda_t)\psi''(\lambda_t))^2}{(\psi''(\lambda_t))^3} \le \frac{1}{16^t}\cdot\frac{(\psi'(\lambda_0)\psi''(\lambda_0))^2}{(\psi''(\lambda_t))^3}.$$

It remains to estimate $\psi'(\lambda_0)$, $\psi''(\lambda_0)$, and $\psi''(\lambda_t)$. Given $\eta p \in [-1,0]^d$ and $\lambda_0 + \eta g(i) \ge 1$, through direct calculation,

$$|\psi'(\lambda_0)| = \sum_{i=1}^d \frac{1 - \eta p(i)}{\lambda_0 + \eta g(i)} - 1 \le \sum_{i=1}^d (1 - \eta p(i)) - 1 < 2d,$$

$$\psi''(\lambda_0) = \sum_{i=1}^d \frac{1 - \eta p(i)}{(\lambda_0 + \eta g(i))^2} \le \sum_{i=1}^d \frac{2}{1} = 2d.$$

Since

$$\psi'(\lambda_t) = 1 - \sum_{i=1}^d \frac{1 - \eta p(i)}{\lambda_t + \eta g(i)} \le 0,$$

by the Cauchy-Schwarz inequality,

$$\psi''(\lambda_t) = \sum_{i=1}^d \frac{1 - \eta p(i)}{(\lambda_t + \eta g(i))^2} \ge \frac{\left(\sum_{i=1}^d \frac{1-\eta p(i)}{\lambda_t + \eta g(i)}\right)^2}{\sum_{i=1}^d (1 - \eta p(i))} \ge \frac{1}{\sum_{i=1}^d (1 - \eta p(i))} \ge \frac{1}{2d}.$$

Therefore, for $0 \le t \le T+1$,

$$\delta(\lambda_t)^2 < \frac{(2d)^7}{16^t}.$$

In particular, $\delta(\lambda_{T+1}) < 1/2$, i.e., $\lambda_{T+1} \in \mathcal{Q}$. This concludes the proof. $\qquad\square$

Proposition F.3 suggests that after $O(\log d)$ iterations of Newton's method, we can switch to the intermediate Newton's method which exhibits quadratic convergence [27, Theorem 5.2.2]. Each iteration of both Newton's method and the intermediate Newton's method takes $O(d)$ time. Therefore, it takes $\tilde{O}(d \log d)$ time to approximate $\lambda^\star$. Given $\lambda^\star$, both $x_{t+1}$ and $\hat{g}_{t+1}$ can be computed in $O(d)$ time, according to their respective definitions in Equation (14). Therefore, the overall time complexity of computing $x_{t+1}$ and $\hat{g}_{t+1}$ is $\tilde{O}(d)$.

### F.2 Proof of Corollary 5.2

By the definition of local norms (3), we write

$$\|g_t - \hat{g}_t + u_t\|_{x_t,*} = \|(g_t + u_t) \odot x_t - p_t \oslash x_t \odot x_t\|_2 = \|(g_t + u_t) \odot x_t - p_t\|_2.$$

By Lemma 4.4 and Theorem 3.2, if $\eta_{t-1}\|(g_t + u_t) \odot x_t - p_t\|_2 \leq 1/2$, then Algorithm 2 satisfies

$$R_T(x) \leq \frac{h(x)}{\eta_T} + \sum_{t=1}^{T} \eta_{t-1}\|(g_t + u_t) \odot x_t - p_t\|_2^2.$$

For any $x \in \Delta$, define $x' = (1 - 1/T)x + e/(dT)$. Then,

$$h(x') \leq -d \log d - \sum_{i=1}^{d} \log\left(\frac{1}{dT}\right) = d \log T, \quad \forall x \in \Delta$$

and hence [22, Lemma 10]

$$R_T(x) \leq R_T(x') + 2 \leq \frac{d \log T}{\eta_T} + \sum_{t=1}^{T} \eta_{t-1}\|(g_t + u_t) \odot x_t - p_t\|_2^2 + 2.$$

## G  Proof of Theorem 6.1 (Gradual-Variation Bound)

### G.1  Proof of Theorem 6.1

We will use the following lemma, whose proof is postponed to the end of the section.

**Lemma G.1.** *Let* $\{x_t\}$ *be the iterates of Algorithm 2. If for all* $t \geq 1$,

$$\eta_t \leq \frac{1}{6}, \quad \eta_{t-1}\|g_t - \hat{g}_t\|_{x_t,*} \leq \frac{1}{6}, \quad \left(1 - \frac{1}{6\sqrt{d}}\right)\eta_{t-1} \leq \eta_t \leq \eta_{t-1}, \quad \text{and} \quad \|\hat{g}_t\|_{x_t,*} \leq 1,$$

(16)

*then* $\|x_{t+1} - x_t\|_{x_t} \leq 1$.

We start with checking the conditions in Lemma G.1.

- It is obvious that $\eta_t \leq 1/(16\sqrt{2}) \leq 1/6$.

- By the definition of the dual local norm (3), the definition of $\hat{g}_t$, and Lemma 4.3,

$$\|\hat{g}_t\|_{x_t,*} = \|\hat{g}_t \odot x_t\|_2 = \|g_{t-1} \odot x_{t-1}\|_2 = \|g_{t-1}\|_{x_{t-1},*} \leq 1, \quad \forall t \geq 2.$$

- Then, by the triangle inequality and Lemma 4.3,

$$\eta_{t-1}\|g_t - \hat{g}_t\|_{x_t,*} \leq \frac{1}{16\sqrt{2}}\left(\|g_t\|_{x_t,*} + \|\hat{g}_t\|_{x_t,*}\right) \leq \frac{1}{8\sqrt{2}} \leq \frac{1}{6}.$$

- It is obvious that the sequence $\{\eta_t\}$ is non-increasing. By the triangle inequality, Lemma 4.3 and the fact that $\sqrt{a+b} \le \sqrt{a} + \sqrt{b}$ for non-negative numbers $a$ and $b$, we write

$$\frac{\eta_t}{\eta_{t-1}} = \frac{\sqrt{512d + 2 + V_{t-1}}}{\sqrt{512d + 2 + V_t}}$$

$$\ge \frac{\sqrt{512d + 2 + V_{t-1}}}{\sqrt{512d + 2 + V_{t-1} + 2}}$$

$$\ge \frac{\sqrt{512d + 2 + V_{t-1}}}{\sqrt{512d + 2 + V_{t-1}} + \sqrt{2}}$$

$$= 1 - \frac{\sqrt{2}}{\sqrt{512d + 2 + V_{t-1}} + \sqrt{2}}$$

$$\ge 1 - \frac{1}{16\sqrt{d}}$$

$$\ge 1 - \frac{1}{6\sqrt{d}}.$$

Then, Lemma G.1 implies that

$$r_t := \|x_{t+1} - x_t\|_{x_t} \le 1.$$

By Proposition C.3,

$$\omega(\|x_t - x_{t+1}\|_{x_t}) \ge \frac{r_t^2}{4}.$$

By Corollary 5.2,

$$R_T \le \frac{d \log T}{\eta_T} + \sum_{t=1}^{T} \left( \langle g_t - p_t \oslash x_t, x_t - x_{t+1} \rangle - \frac{1}{4\eta_{t-1}} r_t^2 \right) + 2.$$

For $t \ge 2$, by Hölder's inequality and the definition of the dual local norm (3),

$$\langle g_t - p_t \oslash x_t, x_t - x_{t+1} \rangle \le \|g_t - p_t \oslash x_t\|_{x_t,*} r_t$$

$$= \|x_t \odot g_t - p_t\|_2 r_t$$

$$= \|x_t \odot g_t - x_{t-1} \odot g_{t-1}\|_2 r_t.$$

By the triangle inequality,

$$\|x_t \odot g_t - x_{t-1} \odot g_{t-1}\|_2 r_t$$
$$\le \|x_t \odot g_t - x_{t-1} \odot \nabla f_t(x_{t-1})\|_2 r_t + \|x_{t-1} \odot \nabla f_t(x_{t-1}) - x_{t-1} \odot g_{t-1}\|_2 r_t.$$

We bound the two terms separately. By the AM-GM inequality and Lemma 4.6, the first term is bounded by

$$\|x_t \odot g_t - x_{t-1} \odot \nabla f_t(x_{t-1})\|_2 r_t \le 4\eta_{t-1} \|x_t \odot g_t - x_{t-1} \odot \nabla f_t(x_{t-1})\|_2^2 + \frac{1}{16\eta_{t-1}} r_t^2$$

$$\le 64\eta_{t-1} r_{t-1}^2 + \frac{1}{16\eta_{t-1}} r_t^2.$$

By the AM-GM inequality, the second term is bounded by

$$\|x_{t-1} \odot \nabla f_t(x_{t-1}) - x_{t-1} \odot g_{t-1}\|_2 r_t = \|\nabla f_t(x_{t-1}) - g_{t-1}\|_{x_{t-1},*} r_t$$

$$\le 4\eta_{t-1} \|\nabla f_t(x_{t-1}) - g_{t-1}\|_{x_{t-1},*}^2 + \frac{1}{16\eta_{t-1}} r_t^2.$$

Combine all inequalities above. We obtain

$$\langle g_t - p_t \oslash x_t, x_t - x_{t+1} \rangle - \frac{1}{4\eta_{t-1}} r_t^2$$

$$\le 4\eta_{t-1} \|\nabla f_t(x_{t-1}) - g_{t-1}\|_{x_{t-1},*}^2 + 64\eta_{t-1} r_{t-1}^2 - \frac{1}{8\eta_{t-1}} r_t^2$$

(17)

For $t = 1$, by a similar argument,

$$\langle g_1, x_1 - x_2 \rangle - \frac{1}{4\eta_0} r_1^2 \leq r_1 - \frac{1}{4\eta_0} r_1^2 \leq \frac{1}{2}\left(4\eta_0 + \frac{1}{4\eta_0} r_1^2\right) - \frac{1}{4\eta_0} r_1^2 = 2\eta_0 - \frac{1}{8\eta_0} r_1^2. \qquad (18)$$

By combining (17) and (18) and noticing that $\eta_{t-1}\eta_t \leq 1/512$,

$$\sum_{t=1}^{T} \langle g_t - p_t \oslash x_t, x_t - x_{t+1} \rangle - \frac{1}{4\eta_{t-1}} r_t^2$$

$$\leq 2\eta_0 + \sum_{t=2}^{T} 4\eta_{t-1}\|\nabla f_t(x_{t-1}) - g_{t-1}\|_{x_{t-1},*}^2 + \sum_{t=1}^{T-1}\left(64\eta_t - \frac{1}{8\eta_{t-1}}\right) r_t^2$$

$$\leq 2\eta_0 + \sum_{t=2}^{T} 4\eta_{t-1}\|\nabla f_t(x_{t-1}) - g_{t-1}\|_{x_{t-1},*}^2.$$

By Corollary 5.2,

$$R_T \leq \frac{d\log T}{\eta_T} + 2\eta_0 + \sum_{t=2}^{T} 4\eta_{t-1}\|\nabla f_t(x_{t-1}) - \nabla f_{t-1}(x_{t-1})\|_{x_{t-1},*}^2 + 2.$$

Plug in the choice of learning rates into the regret bound. We obtain [28, Lemma 4.13]

$$R_T \leq \sqrt{d}\log T\sqrt{V_T + 512d + 2} + 4\sqrt{d}\sum_{t=2}^{T} \frac{\|\nabla f_t(x_{t-1}) - \nabla f_{t-1}(x_{t-1})\|_{x_{t-1},*}^2}{\sqrt{512d + 2 + V_{t-1}}} + 3$$

$$\leq \sqrt{d}\log T\sqrt{V_T + 512d + 2} + 4\sqrt{d}\sum_{t=2}^{T} \frac{\|\nabla f_t(x_{t-1}) - \nabla f_{t-1}(x_{t-1})\|_{x_{t-1},*}^2}{\sqrt{512d + V_t}} + 3$$

$$\leq \sqrt{d}\log T(\sqrt{V_T + 512d} + \sqrt{2}) + 4\sqrt{d}\int_0^{V_T} \frac{\mathrm{d}s}{\sqrt{512d + s}} + 3$$

$$\leq (\log T + 8)\sqrt{dV_T + 512d^2} + \sqrt{2d}\log T + 2 - 128\sqrt{2d}.$$

## G.2 Proof of Lemma G.1

Define

$$y_t \in \arg\min_{x \in \Delta} \langle g_{1:t}, x \rangle + \frac{1}{\eta_{t-1}} h(x), \quad z_t \in \arg\min_{x \in \Delta} \langle g_{1:t}, x \rangle + \frac{1}{\eta_t} h(x).$$

By the triangle inequality,

$$\|x_t - x_{t+1}\|_{x_t} \leq \|x_t - y_t\|_{x_t} + \|y_t - z_t\|_{x_t} + \|z_t - x_{t+1}\|_{x_t}. \qquad (19)$$

Define $d_1 := \|x_t - y_t\|_{x_t}$, $d_2 := \|y_t - z_t\|_{y_t}$, and $d_3 := \|z_t - x_{t+1}\|_{x_{t+1}}$. Suppose that $d_i \leq 1/5$ for all $1 \leq i \leq 3$, which we will verify later. By Theorem C.1,

$$(1 - d_1)^2 \nabla^2 h(x_t) \leq \nabla^2 h(y_t) \leq \frac{1}{(1 - d_1)^2}\nabla^2 h(x_t),$$

$$(1 - d_2)^2 \nabla^2 h(y_t) \leq \nabla^2 h(z_t) \leq \frac{1}{(1 - d_2)^2}\nabla^2 h(y_t), \qquad (20)$$

$$(1 - d_3)^2 \nabla^2 h(x_{t+1}) \leq \nabla^2 h(z_t) \leq \frac{1}{(1 - d_3)^2}\nabla^2 h(x_{t+1}).$$

We obtain

$$\|y_t - z_t\|_{x_t} \leq \frac{d_2}{1 - d_1}, \quad \|z_t - x_{t+1}\|_{x_t} \leq \frac{d_3}{(1 - d_1)(1 - d_2)(1 - d_3)}.$$

Then, the inequality (19) becomes

$$\|x_t - x_{t+1}\|_{x_t} \leq d_1 + \frac{d_2}{1 - d_1} + \frac{d_3}{(1 - d_1)(1 - d_2)(1 - d_3)}.$$

It is easily checked that the maximum of the right-hand side is attained at $d_1 = d_2 = d_3 = 1/5$ and the maximum value equals $269/320$. This proves the lemma.

Now, we prove that $d_i \leq 1/5$ for all $1 \leq i \leq 3$.

1. (Upper bound of $d_1$.) By the optimality conditions of $x_t$ and $y_t$,

$$\langle \eta_{t-1} g_{1:t} + \nabla h(y_t), x_t - y_t \rangle \geq 0$$
$$\langle \eta_{t-1} g_{1:t-1} + \eta_{t-1} \hat{g}_t + \nabla h(x_t), y_t - x_t \rangle \geq 0.$$

Sum up the two inequalities. We get

$$\eta_{t-1} \langle g_t - \hat{g}_t, x_t - y_t \rangle \geq \langle \nabla h(x_t) - \nabla h(y_t), x_t - y_t \rangle$$

By Hölder's inequality, Lemma 4.4, and Theorem C.2, we obtain

$$\eta_{t-1} \| g_t - \hat{g}_t \|_{x_t,*} d_1 \geq \frac{d_1^2}{1 + d_1}.$$

Then, by the assumption (16), we write

$$d_1 \leq \frac{\eta_{t-1} \| g_t - \hat{g}_t \|_{x_t,*}}{1 - \eta_{t-1} \| g_t - \hat{g}_t \|_{x_t,*}} \leq 1/5.$$

2. (Upper bound of $d_2$.) By the optimality conditions of $y_t$ and $z_t$,

$$\langle g_{1:t} + \frac{1}{\eta_t} \nabla h(z_t), y_t - z_t \rangle \geq 0$$

$$\langle g_{1:t} + \frac{1}{\eta_{t-1}} \nabla h(y_t), z_t - y_t \rangle \geq 0.$$

Sum up the two inequalities. We get

$$\left( \frac{1}{\eta_t} - \frac{1}{\eta_{t-1}} \right) \langle \nabla h(y_t), y_t - z_t \rangle \geq \frac{1}{\eta_t} \langle \nabla h(y_t) - \nabla h(z_t), y_t - z_t \rangle.$$

By Definition C.4, Lemma C.5, and Theorem C.2,

$$\left( \frac{1}{\eta_t} - \frac{1}{\eta_{t-1}} \right) \sqrt{d} d_2 \geq \frac{1}{\eta_t} \frac{d_2^2}{1 + d_2}.$$

Then, by the assumption (16), we write

$$d_2 \leq \frac{(1 - \eta_t / \eta_{t-1}) \sqrt{d}}{1 - (1 - \eta_t / \eta_{t-1}) \sqrt{d}} \leq \frac{1}{5}.$$

3. (Upper bound of $d_3$.) By the optimality conditions of $z_t$ and $x_{t+1}$,

$$\langle \eta_t g_{1:t} + \eta_t \hat{g}_{t+1} + \nabla h(x_{t+1}), z_t - x_{t+1} \rangle \geq 0$$
$$\langle \eta_t g_{1:t} + \nabla h(z_t), x_{t+1} - z_t \rangle \geq 0.$$

Sum up the two inequalities. We get

$$\eta_t \langle \hat{g}_{t+1}, z_t - x_{t+1} \rangle \geq \langle \nabla h(z_t) - \nabla h(x_{t+1}), z_t - x_{t+1} \rangle.$$

By Hölder's inequality, the assumption (16), and Theorem C.2, we obtain

$$\eta_t d_3 \geq \eta_t \| \hat{g}_{t+1} \|_{x_{t+1},*} d_3 \geq \frac{d_3^2}{1 + d_3}.$$

Then, by the assumption (16), we write

$$d_3 \leq \frac{\eta_t}{1 - \eta_t} \leq 1/5.$$

# H   Proof of Theorem 6.2 (Small-Loss Bound)

Recall that $\alpha_t$ minimizes $\|g_t + \alpha e\|_{x_t,*}^2$ over all $\alpha \in \mathbb{R}$ (Remark 4.8). We write

$$\|g_t + \alpha_t e\|_{x_t,*}^2 \leq \|g_t\|_{x_t,*}^2 \leq 1,$$

which, with the observation that $\eta_t \leq 1/2$ for all $t \in \mathbb{N}$, implies that $\eta_{t-1}\|g_t + \alpha_t e\|_{x_t,*} \leq 1/2$. Then, since the all-ones vector $e$ is orthogonal to the set $\Delta - \Delta$, by Corollary 5.2 with $p_t = 0$ and $u_t = \alpha_t e$, we get

$$R_T \leq \frac{d \log T}{\eta_T} + \sum_{t=1}^{T} \eta_{t-1}\|(g_t + \alpha_t e) \odot x_t\|_2^2 + 2$$

$$= \frac{d \log T}{\eta_T} + \sum_{t=1}^{T} \eta_{t-1}\|g_t + \alpha_t e\|_{x_t,*}^2 + 2$$

Plug in the explicit expressions of the learning rates $\eta_t$. We write [28, Lemma 4.13]

$$R_T \leq \frac{d \log T}{\eta_T} + \sqrt{d} \sum_{t=1}^{T} \frac{\|g_t + \alpha_t e\|_{x_t,*}^2}{\sqrt{4d + 1 + \sum_{\tau=1}^{t-1}\|g_\tau + \alpha_\tau e\|_{x_\tau,*}^2}} + 2$$

$$\leq \frac{d \log T}{\eta_T} + \sqrt{d} \sum_{t=1}^{T} \frac{\|g_t + \alpha_t e\|_{x_t,*}^2}{\sqrt{4d + \sum_{\tau=1}^{t}\|g_\tau + \alpha_\tau e\|_{x_\tau,*}^2}} + 2$$

$$\leq \sqrt{d} \log T \sqrt{4d + 1 + \sum_{t=1}^{T}\|g_t + \alpha_t e\|_{x_t,*}^2} + \sqrt{d} \int_0^{\sum_{t=1}^{T}\|g_t + \alpha_t e\|_{x_t,*}^2} \frac{\mathrm{d}s}{\sqrt{4d + s}} + 2$$

$$\leq (\log T + 2)\sqrt{d \sum_{t=1}^{T}\|g_t + \alpha_t e\|_{x_t,*}^2 + 4d^2 + d}$$

$$\leq (\log T + 2)\sqrt{d \sum_{t=1}^{T}\|g_t + \alpha_t e\|_{x_t,*}^2 + 4d^2 + d}.$$

Finally, by Lemma 4.7,

$$R_T \leq (\log T + 2)\sqrt{4d \sum_{t=1}^{T} f_t(x_t) + 4d^2 + d}.$$

The theorem then follows from Lemma 4.23 of Orabona [28].

# I   A Second-Order Regret Bound

In this section, we introduce AA + LB-FTRL with Average-Multiplicative-Gradient Optimism (Algorithm 6) that achieves a second-order regret bound. This bound is characterized by the variance-like quantity

$$Q_T := \min_{p \in \mathbb{R}^d} \sum_{t=1}^{T} \|x_t \odot \nabla f_t(x_t) - p\|_2^2.$$

For comparison, existing second-order bounds are characterized by the quantity

$$V_T := \min_{u \in \mathbb{R}^d} \sum_{t=1}^{T} \|\nabla f_t(x_t) - u\|_2^2,$$

$T$ times the empirical variance of the gradients (see, e.g., Section 7.12.1 in the lecture note of Orabona [28]). Given that every $p \in \mathbb{R}^d$ can be expressed as $p = x_t \odot v$ for $v = p \oslash x_t$, and by the definition of the dual local norm (3), we obtain

$$Q_T = \min_{v \in \mathbb{R}^d} \sum_{t=1}^{T} \|\nabla f_t(x_t) - v\|_{x_t,*}^2,$$

showing that $Q_T$ is a local-norm analogue of $V_T$.

The flexibility introduced by optimizing $Q_T$ over all $v \in \mathbb{R}^d$ allows for a fast regret rate when the data is easy. In particular, Theorem I.2 in Section I.3 shows that when $Q_T \leq 1$, then Algorithm 6 achieves a fast $O(d \log T)$ regret rate. However, while $Q_T$ looks reasonable as a mathematical extension of $V_T$, the empirical variance interpretation for $V_T$ does not extend to $Q_T$. Interpreting $Q_T$ or seeking a "more natural" variant of $Q_T$ is left as a future research topic.

## I.1 The Gradient Estimation Problem

To obtain a better regret bound by using Corollary 5.2, one has to design a good strategy of choosing $p_t$. In Section 6.1, we have used $p_t = x_{t-1} \odot g_{t-1}$. In this section, we consider a different approach.

Consider the following online learning problem, which we call the *Gradient Estimation Problem*. It is a multi-round game between LEARNER and REALITY. In the $t$-th round, LEARNER chooses a vector $p_t \in \mathbb{R}^d$; then, REALITY announces a convex loss function $\ell_t(p) = \eta_{t-1}\|x_t \odot g_t - p\|_2^2$; finally, LEARNER suffers a loss $\ell_t(p_t)$.

## I.2 LB-FTRL with Average-Multiplicative-Gradient Optimism

In the gradient estimation problem, the loss functions are strongly-convex. A natural strategy of choosing $p_t$ is then following the leader (FTL). In the $t$-th round, FTL suggests choosing

$$p_t = \frac{1}{\eta_{0:t-2}} \sum_{\tau=1}^{t-1} \eta_{\tau-1} x_\tau \odot g_\tau \in \arg\min_{p \in \mathbb{R}^d} \sum_{\tau=1}^{t-1} \ell_\tau(p).$$

This strategy leads to Algorithm 5. Theorem I.1 provides a regret bound for Algorithm 5.

---

**Algorithm 5** LB-FTRL with Average-Multiplicative-Gradient Optimism

**Input:** A sequence of learning rates $\{\eta_t\}_{t \geq 0} \subseteq \mathbb{R}_{++}$.

1: $h(x) := -d \log d - \sum_{i=1}^{d} \log x(i)$.
2: $x_1 \leftarrow \arg\min_{x \in \Delta} \eta_0^{-1} h(x)$.
3: **for all** $t \in \mathbb{N}$ **do**
4:     Announce $x_t$ and receive $a_t$.
5:     $g_t := \nabla f_t(x_t)$.
6:     $p_{t+1} \leftarrow (1/\eta_{0:t-1}) \sum_{\tau=1}^{t} \eta_{\tau-1} x_\tau \odot g_\tau$.
7:     Solve $x_{t+1}$ and $\hat{g}_{t+1}$ satisfying

$$\begin{cases} x_{t+1} \odot \hat{g}_{t+1} = p_{t+1}, \\ x_{t+1} \in \arg\min_{x \in \Delta} \langle g_{1:t}, x \rangle + \langle \hat{g}_{t+1}, x \rangle + \eta_t^{-1} h(x). \end{cases}$$

8: **end for**

---

**Theorem I.1.** *Assume that the sequence of learning rates $\{\eta_t\}$ is non-increasing and $\eta_0 \leq 1/(2\sqrt{2})$. Then, Algorithm 5 achieves*

$$R_T \leq \frac{d \log T}{\eta_T} + 2 \sum_{t=1}^{T} \frac{\eta_{t-1}^2}{\eta_{0:t-1}} + 2 + \min_{p \in \mathbb{R}^d} \sum_{t=1}^{T} \eta_{t-1}\|x_t \odot \nabla f_t(x_t) - p\|_2^2.$$

*If $\eta_t = \eta \leq 1/(2\sqrt{2})$ is a constant, then*

$$R_T \leq \frac{d \log T}{\eta} + 2\eta(\log T + 1) + 2 + \eta Q_T.$$

*Proof.* Since $p_t$ is a convex combination of $x_\tau \odot g_\tau$, by Lemma 4.1, $p_t \in -\Delta$. We restrict the action set of the gradient estimation problem to be $-\Delta$. By the regret bound of FTL for strongly convex losses [28, Corollary 7.24],

$$\sum_{t=1}^{T} \ell_t(p_t) - \min_{p \in -\Delta} \sum_{t=1}^{T} \ell_t(p) \leq \frac{1}{2} \sum_{t=1}^{T} \frac{(2\sqrt{2}\eta_{t-1})^2}{2\eta_{0:t-1}} = 2 \sum_{t=1}^{T} \frac{\eta_{t-1}^2}{\eta_{0:t-1}}.$$

Then, by Corollary 5.2,

$$R_T \leq \frac{d \log T}{\eta_T} + 2 + \sum_{t=1}^{T} \ell_t(p_t) \leq \frac{d \log T}{\eta_T} + 2 + 2 \sum_{t=1}^{T} \frac{\eta_{t-1}^2}{\eta_{0:t-1}} + \min_{p \in -\Delta} \sum_{t=1}^{T} \ell_t(p).$$

The first bound in the theorem follows from

$$\min_{p \in -\Delta} \sum_{t=1}^{T} \ell_t(p) = \min_{p \in \mathbb{R}^d} \sum_{t=1}^{T} \eta_{t-1} \|x_t \odot g_t - p\|_2^2.$$

The second bound in the theorem follows from the inequality $\sum_{t=1}^{T} \eta^2/(t\eta) \leq \eta(\log T + 1)$. $\quad\square$

**Time Complexity.** Suppose that a constant learning rate is used. Then, both $g_t$ and $p_{t+1}$ can be computed in $O(d)$ arithmetic operations. By Theorem 5.1, $x_{t+1}$ can be computed in $\tilde{O}(d)$ time. Hence, the per-round time of Algorithm 5 is $\tilde{O}(d)$.

### I.3  AA + LB-FTRL with Average-Multiplicative-Gradient Optimism

The constant learning rate that minimizes the regret bound in Theorem I.1 is $\eta = O(\sqrt{d \log T/Q_T})$. However, the value of $Q_T$ is not unknown to INVESTOR initially.

In this section, we propose Algorithm 6, which satisfies a second-order regret bound without the need for knowing $Q_T$ in advance. Algorithm 6 uses multiple instances of Algorithm 5, which we call *experts*, with different learning rates. The outputs of the experts are aggregated by the Aggregating Algorithm (AA) due to Vovk [36].

---

**Algorithm 6** AA + LB-FTRL with Average-Multiplicative-Gradient Optimism

**Input:** The time horizon $T$.
1: $K := 1 + \lceil \log_2 T \rceil$.
2: **for all** $k \in [K]$ **do**
3: $\quad w_1^{(k)} \leftarrow 1$.
4: $\quad q^{(k)} \leftarrow 2^k$.
5: $\quad \eta^{(k)} \leftarrow \frac{\sqrt{d \log T}}{2\sqrt{2d \log T} + \sqrt{q^{(k)}}}$.
6: $\quad$ Initialize Algorithm 5 with the constant learning rate $\eta^{(k)}$ as the $k$-th expert $\mathcal{A}_k$.
7: $\quad$ Obtain $x_1^{(k)}$ from $\mathcal{A}_k$.
8: **end for**
9: $x_1 := \frac{1}{\sum_{k=1}^{K} w_1^{(k)}} \sum_{k=1}^{K} w_1^{(k)} x_1^{(k)}$.
10: **for all** $t \in [T]$ **do**
11: $\quad$ Observe $a_t$.
12: $\quad$ **for all** $k \in [K]$ **do**
13: $\quad\quad w_{t+1}^{(k)} = w_t^{(k)} \langle a_t, x_t^{(k)} \rangle$.
14: $\quad\quad$ Feed $a_t$ into $\mathcal{A}_k$ and obtain $x_{t+1}^{(k)}$.
15: $\quad$ **end for**
16: $\quad$ Output $x_{t+1} := \frac{1}{\sum_{k=1}^{K} w_{t+1}^{(k)}} \sum_{k=1}^{K} w_{t+1}^{(k)} x_{t+1}^{(k)}$.
17: **end for**

---

The regret bound of Algorithm 6 is stated in Theorem I.2, which follows from the regret of AA and Theorem I.1.

**Theorem I.2.** *Algorithm 6 with input $T \in \mathbb{N}$ satisfies*

$$R_T \leq \begin{cases} (1+\sqrt{2})\sqrt{dQ_T \log T} + 2\sqrt{2}d\log T + \frac{1}{\sqrt{2}}\log T + \log(\log_2 T + 2) + 3, & \text{if } Q_T \geq 1, \\ 2\sqrt{2}d\log T + \frac{1}{\sqrt{2}}\log T + \sqrt{2d\log T} + \log(\log_2 T + 2) + 4, & \text{if } Q_T < 1. \end{cases}$$

**Remark I.3.** *Since $Q_T \leq 2T$, the regret bound in Theorem I.2 is $O(\sqrt{dT\log T})$ in the worst case.*

*Proof.* $K = 1 + \lceil \log_2 T \rceil$ is the number of experts in Algorithm 6. Let $R_T^{(k)}$ be the regret of the $k$-th expert $\mathcal{A}_k$. Since the loss functions $f_t(x) = -\log \langle a_t, x \rangle$ are 1-mixable, by the regret bound of AA [36],

$$R_T \leq \min_{k \in [K]} R_T^{(k)} + \log K,$$

where $R_T$ and $R_T^{(k)}$ denote the regret of Algorithm 6 and the regret of the $k$-th expert, respectively.

The quantity $q^{(k)}$ in Algorithm 6 is used as an estimate of $Q_T$. Notice that $Q_T \leq 2T \leq q^{(K)}$. If $Q_T \geq 1$, then there exists $k^\star \in [K]$ such that $q^{(k^\star - 1)} \leq Q_T \leq q^{(k^\star)}$. By Theorem I.1 and some direct calculations,

$$R_T^{(k^\star)} \leq (1+\sqrt{2})\sqrt{dQ_T \log T} + 2\sqrt{2}d\log T + \frac{1}{\sqrt{2}}\log T + 3.$$

If $Q_T < 1$, then by Theorem I.1,

$$R_T^{(1)}(x) \leq 2\sqrt{2}d\log T + \sqrt{2d\log T} + \frac{1}{\sqrt{2}}\log T + 4.$$

The theorem follows by combining the inequalities above. $\qquad \square$

**Time Complexity.**   In each round, Algorithm 6 requires implementing $O(\log\log T)$ copies of Algorithm 5. By Theorem 5.1, the per-round time is $\tilde{O}(d\log\log T)$.

