# OpenReview forum: "Data-Dependent Bounds for Online Portfolio Selection Without Lipschitzness and Smoothness"
_NeurIPS.cc/2023/Conference — NeurIPS 2023 poster_

### Official Review · Reviewer_Xwzr · 2023-07-04

**Soundness:** 4 excellent
**Presentation:** 3 good
**Contribution:** 3 good
**Rating:** 7
**Confidence:** 2

**Summary:**

The paper considers the online portfolio selection problem. In this problem, one must allocate funds between d possible investment choices, with the goal of maximizing the total amount. In each round, the "success" of each choice is revealed, in the form of a ratio between new and old price, called price relative; these price relatives are adversarially chosen. The goal is to have low regret, in terms of the logarithm of total wealth, against any constant, fractional allocation of funds between the investment choices.

In general, where the number of rounds is T, existing work for this problem exhibits either polynomial running time in T and poly-logarithmic regret in T, or running time independent of T and regret polynomial in T (specifically, square root).

The paper considers a different approach, providing data-dependent bounds for the problem.
The paper presents three bounds w.r.t. the input: small-loss bound, w.r.t. the total loss of the optimal allocation; gradual-variation bound, w.r.t. the volatility of the gradients of the loss functions; and a second-order bound, w.r.t. some second-order statistics of the loss function. The first two appear in the body of the paper.

The paper presents an algorithm based on optimistic FTRL w.r.t. a log-barrier regularizer; the optimism refers to using an estimate for the upcoming loss function in choosing the allocation for the next time step. To my understanding, the algorithmic contribution of the paper is in the method for choosing the loss estimate.

Comments:
Line 33: Maybe "lower" instead of "faster"
Line 283: In the displayed equation, should be a_t instead of a


**Strengths:**

The online portfolio selection is an interesting problem, and obtaining data-dependent bounds for this problem seems natural.
In that sense, the paper does a comprehensive job and provides bounds w.r.t. three parameters.

**Weaknesses:**

I have a concern regarding the small-loss bound, addressed in the questions for rebuttal.

**Questions:**

Regarding Assumption 1, the paper correctly states that this could be ensured by normalizing the price-relatives vector, without affecting regret. However, this does affect the total loss of the adversary, which comes into play in the small-loss bound in Theorem 6.2. If I understand correctly, the assumption is thus nontrivial to make, which weakens Theorem 6.2. Is this the case?

**Limitations:**

yes

---

> ### Author Rebuttal · Authors · 2023-08-08
>
> Thanks for your careful review.
> 1. **Typos:** We will correct them. Thank you.
> 2. **Concern on the small-loss bound:** If the price relatives are not upper-bounded by 1, then the cumulative loss in the small loss bound is defined with respect to the normalized price relatives. Hence, the assumption that the price relatives are upper-bounded by 1 does not weaken the regret bound. Thank you for pointing out this potential confusion. We did not notice that because of the notations, the remark following Assumption 1 is not sufficient. We will clarify this after Theorem 6.2 in the revision.

---

> > ### Comment · Reviewer_Xwzr · 2023-08-15
> >
> > Okay, thank you for clearing up this issue. I am raising my score from 6 to 7.

---

### Official Review · Reviewer_q9Qg · 2023-07-05

**Soundness:** 3 good
**Presentation:** 3 good
**Contribution:** 3 good
**Rating:** 6
**Confidence:** 4

**Summary:**

This work studies online portfolio selection (OPS) problem and establishes regret bounds that is square-root dependent on some data-dependent quantities, namely, the cumulative loss of the best action or the variation of the gradients respectively, without lipschitzness or smoothness assumption. Although previous studies obtain regret bounds with logarithmic dependence on the same data-dependent factors, they require the lipschitzness assumptions, which this work circumvents by investigating the problem carefully and proposing alternative local-norm lemmas as substitutes for lipschitzness and smoothness conditions. Furthermore, this work designs a novel optimistic FTRL algorithm with a self-concordant function to cooperate with the local-norm techniques and validates that the prediction can be resolved in $\tilde{O}(d)$ time pre round.

**Strengths:**

1.	This paper leverages the inherent structure of the problem and derives the first data-dependent regret bounds without lipschitzness or smoothness assumptions for the OPS setting.
2.	The proposed optimistic FTRL algorithm is novel and interesting, which facilitates the application of local-norm techniques with optimism for this problem.
3.	This paper is clearly written and presented well.



**Weaknesses:**

The OPS problem is fundamentally challenging, and I am glad to witness advancements in this field even if the results have not yet reached an optimal level. Below are some questions that emerged when I am reviewing the paper, and I would be appreciated if the authors can provide some feedback.

1.	Ordinarily, one might anticipate that data-dependent bounds would demonstrate a marked advantage over minimax optimal results in the best-case scenarios, such as when $V_T = 0$ or $L_T^\star = 0$. But, by Theorem 6.1, when $ V_T = 0$, the algorithm can only assure $O(d\log T)$ regret bound, just matching the minimax optimal bound. By Theorem 6.2, $L_T^\star = 0$ will imply an even less optimal bound than the minimax result. However, these results do attain Pareto frontier optimality when balancing the optimality and efficiency under the best cases. I advise the authors to elaborate further on the motivation and highlight the efficiency of the proposed algorithms to make the contributions clearer.
2.	As the authors noted in line 242, the term "implicit" might potentially mislead readers since it is commonly associated with another class of algorithms [Kulis and Bartlett, 2010]. It may be beneficial to consider a different nomenclature for the algorithm.
3.	I’m interested in the relationship between two data-dependent bounds. Indeed, some earlier study (see Zhao et al., 2021, Theorem 6) posits that a gradual variation dynamic regret can imply a small-loss dynamic regret in the analysis, for general convex, non-negative, and smooth functions. This result is also true for static regret. A natural question that arises from this is whether it's feasible for one algorithm to concurrently secure guarantees for both the gradual-variation bound and small-loss bound.

References:

Brian Kulis and Peter L. Bartlett. Implicit Online Learning. ICML 2010.
Peng Zhao, Yu-Jie Zhang, Lijun Zhang, and Zhi-Hua Zhou. Adaptivity and Non-stationarity: Problem-dependent Dynamic Regret for Online Convex Optimization. ArXiv:2112.14368, 2021.


**Questions:**

See comments above.

---

> ### Author Rebuttal · Authors · 2023-08-08
>
> Thank you for appreciating the online portfolio selection problem and our work.
> 1. **Benefit of data-dependent bounds:** This work is motivated by the high computational complexities of existing logarithmic-regret algorithms. Given that the optimal tradeoff between regret and efficiency remains unclear, we seek the opportunity to attain the optimal regret rate with moderate computational complexity. The results in this paper show there are indeed algorithms that automatically exploit such opportunities. If you agree with this argument, we will make this clearer in the revision.
> 2. **Another name for Algorithm 2:** We agree with the comment. We will rename Algorithm 2 as “LB-FTRL with Multiplicative-Gradient Optimism”. The name is motivated by the fact that, unlike standard optimistic algorithms, we predict $x_t \odot g_t$ instead of the gradient $g_t$ in each round.
> 3. **Possibility of a small-loss gradual-variation bound:** Thanks for pointing out the reference. We had also noticed this work.
>     - Yes, we can construct an algorithm that satisfies a unified small-loss gradual-variation bound. It suffices to aggregate the outputs of Algorithm 3 and Algorithm 4 by Vovk's aggregating algorithm. Since the losses are mixable and there are only two experts, the regret of the resulting algorithm is bounded by $\min \lbrace R_{3, T}, R_{4, T} \rbrace + \log 2$, where $R_{i, T}$ denotes the regret of Algorithm $i$.
>     - The reference by Zhao et al. (2021) that you pointed out provides a more efficient approach for standard smooth losses. Unfortunately, the techniques therein do not directly apply to our case, because our "self-bounding property" (Lemma 4.7) includes an additional $\alpha^\star (x) e$ term. We will mention generalizing Zhao et al. (2021) as a future research direction.
>
> Reference:
> - P. Zhao et al., Adaptivity and non-stationarity: Problem-dependent dynamic regret for online convex optimization, 2021.

---

> ### Comment · Area_Chair_66Gx · 2023-08-16
> **Please acknowledge rebuttal**
>
> Dear reviewer,
>
> Please acknowledge that you have read the rebuttal and indicate whether it adequately addresses your comments. The author-reviewer discussion period ends Aug 21; please engage with the authors before that if needed.
>
> Thanks,
> AC

---

### Official Review · Reviewer_cmFV · 2023-07-06

**Soundness:** 3 good
**Presentation:** 3 good
**Contribution:** 3 good
**Rating:** 7
**Confidence:** 2

**Summary:**

The paper presents beyond-the-worst-case regret bounds for Online Portfolio Selection (OPS). In general online learning, beyond-the-worst-case bounds are established using structural assumptions on the loss functions, such as Lipschitzness and smoothness, but the loss functions in OPS are neither Lipschitz nor smooth, which is the main technical difficulty the paper addresses. To this end, local norm analogues of Lipschitzness and smoothness are established for OPS, and a generic optimistic FTRL algorithm with the log-barrier regularizer is proposed. Specializations of this algorithm achieve a gradient-variation-dependent bound and a $L^*$ bound, which are the first without the additional no junk bond assumption. In the worst case, these bounds match a type of classical regret-computation tradeoff. In better cases, these bounds are logarithmic in $T$, which is a substantial acceleration.

**Strengths:**

- Online portfolio selection is an iconic problem in online learning, with plenty of recent progresses in its worst case characterization. The paper introduces classical types of data dependent bounds to this problem, which has a very clear and natural motivation.

- The challenge of non-Lipschitzness and non-smoothness is technically nontrivial. The solution relies on establishing local counterparts of these properties, which is interesting, and could be of broader applicability.

- The specific data-dependent bounds are novel without the additional no junk bond assumption.

- Related works are discussed in depth, which gives the new contributions a nice context.

**Weaknesses:**

Overall this is a good paper, and there isn't any major criticism I'd like to make. On less important issues,

- The technical presentation of the paper could be improved. Currently there are quite a few of typos and unclear notations. For example, there's Algorithm 1 in the main paper, and an algorithm 1 in the appendix. $\omega$ in Theorem 3.2 is not defined in the main paper. The notation $x(i)$ is used in line 201 but defined in line 227, ...

- I think compared to data-dependent bounds in general online learning, the benefit of data-dependent bounds in OPS is a little less clear, particularly due to the existence of a computationally inefficient logarithmic regret algorithm. I would appreciate more discussions on computation, since the paper essentially considers a particular regret computation tradeoff. Is $\tilde O(d)$ runtime and $\tilde O (\sqrt{dT})$ regret pareto-optimal for OPS in some sense? A numerical example would also be helpful, as it will make the computational savings over universal portfolio (and line 33 to 36) more clear.

**Questions:**

- Could the authors comment a bit more on the novelty of the smoothness characterizations (Lemma 4.6 and 4.7)?

- I'm also generally wondering about the regret computation tradeoff in OPS. Are there lower bounds?

**Limitations:**

The limitations are adequately addressed.

---

> ### Author Rebuttal · Authors · 2023-08-08
>
> Thank you for your appreciation of our work.
> 1. **Novelty of the Smoothness Characterizations:**
>     We are not aware of any similar results in the literature. The closest is perhaps the well-known local smoothness property of self-concordant functions.
> 2. **Optimal regret-efficiency tradeoff:**
>     To the best of our knowledge, there is not a characterization of the optimal tradeoff between regret and computational efficiency for online portfolio selection. Zimmert et al. (2022) consider the currently best tradeoff for *existing* algorithms.
> 3. **Typos:** Thanks for the careful review. We will correct the typos.
> 4. **Benefit of data-dependent bounds:** This work is motivated by the high computational complexities of existing logarithmic-regret algorithms. Given that the optimal tradeoff between regret and efficiency remains unclear, we seek the opportunity to attain the optimal regret rate with moderate computational complexity. The results in this paper show there are indeed algorithms that automatically exploit such opportunities. If you agree with this argument, we will make this clearer in the revision.
>
> Reference:
> - J. Zimmert et al., Pushing the efficiency-regret Pareto frontier for online learning of portfolios and quantum states, 2022.

---

> > ### Comment · Reviewer_cmFV · 2023-08-11
> >
> > I appreciate your rebuttal. It adequately addressed my comments. Although the benefit compared to the computationally inefficient log regret algorithm is still a bit murky, I agree that further studies are beyond the scope of this paper. Within the computationally tractable paradigm, the results are good contributions to the field.

---

> > > ### Author Response · Authors · 2023-08-14
> > > **Thank you**
> > >
> > > Thank you for the reply and appreciation of our work.

---

### Official Review · Reviewer_DgZU · 2023-07-11

**Soundness:** 3 good
**Presentation:** 2 fair
**Contribution:** 3 good
**Rating:** 6
**Confidence:** 2

**Summary:**

The paper studies follow the regularized leader algorithm (FTRL) on the online portfolio selection problem without the assumption of no junk bonds. This makes the resulting loss function non-Lipschitz and non-smooth and makes analyzing the regularized follow the leader algorithm hard to analyze. The paper proposes using a self-concordant regularizer for the FTRL algorithm and proves regret bounds for the algorithm for a generic online convex optimization problem. They then use the result to propose two novel algorithms that have small-loss and gradual-variation regret bounds, respectively, for the online portfolio selection problem, which to the best of the author’s knowledge are the first regret bounds for non-Lipschitz and non-smooth losses.

**Strengths:**

The paper seems significant as it provides a new result towards bounding regret for non-Lipschitz and non-smooth losses. The main contribution seems to originate from being able to prove that FTRL with a self-concordant regularizer without the barrier requirement obtains a regret bound similar to the setting with the barrier requirement. They then show how to apply the result to the online portfolio selection problem which is a canonical online convex optimization problem. The paper provides clear background for the problem to help motivate the goal and significance of the result.

**Weaknesses:**

The paper lacks clarity in how exactly Theorem 3.2 is applied to the online portfolio selection problem. Specifically, it is unclear what conditions are needed to apply Theorem 3.2 as Algorithm 1 is applied to a very specific online convex optimization problem. I believe Section 4 tries to highlight these conditions via Lemma 4.4, 4.5, 4.6, and 4.7, but without looking at the appendix, it is unclear where the conditions factor into proving the regret bounds.

Moreover, in Section 5 where Theorem 3.2 is applied to the online portfolio selection problem, it is very unclear why the implicit optimistic LB-FTRL Algorithm 2 solves it. The first challenge in understanding how Algorithm 2 maps the OPS problem to Algorithm 1 is in the limited explanation in the mapping. From my reading, it seems that the only explanation in the main body is in the second sentence of the first paragraph in Section 5, “By the convexity of the loss functions, OPS can be reduced to an online linear optimization problem described in Section 3 with $v_t = g_t$ and $\mathcal{X}$ being the probability simplex $\Delta$.” Writing the optimization problem explicitly would be more clear.

**Questions:**

Questions
1. Is it possible to state Theorem 3.2 for a generic loss that satisfies some set of specific conditions? This might streamline sections 3 and 4 by highlighting the necessary conditions ahead of time. Additionally, it might reduce the repetition in redefining the problem since fundamentally the only thing that changed was the loss function.
2. Is there intuition how the newly proposed algorithms get around the non-Lipschitz loss? My intuition is that it depends on choosing the learning rate properly so that when the Lipschitz constant is large, the correct learning rate mitigates it. Is the intuition correct? This seems to imply the learning rate should change depending on where the location of the data point for FTL type algorithms.

**Limitations:**

The paper is generally more theory-oriented and authors imply it is unclear how the analysis can be generalized to other settings, thus adequately addressing the limitations.

---

> ### Author Rebuttal · Authors · 2023-08-08
>
> Thank you for the comments.
> 1. **On stating Theorem 3.2 for generic losses:** Yes, it is possible to state Theorem 3.2, which is currently stated for online linear optimization, for generic convex losses. Denote the loss function in the $t$-th round by $f_t$. The only modification required is to replace the loss vectors $v_t$ in Algorithm 1 with the gradients $\nabla f_t ( x_t )$. This follows from the fact that $f_t ( x_t ) - f_t ( x ) \leq \langle \nabla f_t ( x_t ), x_t \rangle - \langle \nabla f_t ( x_t ), x \rangle$ by the convexity of $f_t$, showing that it suffices to solve the online linear optimization problem with $v_t = \nabla f_t ( x_t )$. "Linearizing" the losses is a standard approach. See, e.g., Section 2.4 of Shalev-Shwartz (2012) and Section 2.3 of Orabona (2023).
>
>     We chose to state Theorem 3.2 in the current way to follow the style of Shalev-Shwartz (2012) and Orabona (2023), whereas Hazan (2022) does not explicitly mention linearization and hides that in the proofs. Please let us know if you think it necessary to restate Theorem 3.2 for general loss functions. The modification is easy to do.
> 2. **Intuition on how the non-Lipschitzness issue is solved:** Indeed, as we focus on deriving data-dependent bounds, the issue we face is lack of smoothness rather than lack of Lipschitzness. The lack-of-smoothness issue is addressed by Lemma 4.6 and Lemma 4.7, which provide local-norm analogies of the definition of smoothness and the self-bounding property, respectively.
>
>     Based on our results, one can also obtain a worst-case $\tilde{O} ( \sqrt{ dT } )$ regret bound for online portfolio selection by FTRL with the log-barrier, with a constant learning rate, and without optimisim (a very simple special case of Algorithm 2). The regret bound then follows from Corollary 5.2 and Lemma 4.3. The non-Lipschitzness issue is solved by Lemma 4.3, showing that the gradients are bounded in dual local norms.
>
> References:
> - S. Shalev-Shwartz, Online Learning and Online Convex Optimization, 2012.
> - E. Hazan, Introduction to Online Convex Optimization, 2022.
> - F. Orabona. A Modern Introduction to Online Learning. 2023.

---

### Official Review · Reviewer_mCNh · 2023-07-26

**Soundness:** 4 excellent
**Presentation:** 4 excellent
**Contribution:** 4 excellent
**Rating:** 7
**Confidence:** 4

**Summary:**

This paper studies how to achieve adaptive regret bound, including gradient-variation bound and small-loss bound, for the online portfolio management problem, without the classical no-junk bund assumption. The authors successfully achieve this goal by observing a new kind of smoothness for the function -log wx.

**Strengths:**

The finding of a kind of smoothness of -log<wx> is the major contribution of the paper. I think this is a novel and significant observation, and lead to multiple new conclusions. Specially, for the small-loss bound, one can easily show a upper bound related to root{sum of gradients at x_t} by using a step size inversely proportional to root{sum of norm of gradients}. However, to get a small loss, bound, the next step is to assume smoothness to create a relationship between root{sum of gradients at x_t} and root{sum of function values at x_t} (using the classical Lemma 2.1 of [29]). In this paper, the authors show that, in the OPS problem, for the loss -log<wx>, we do not need smooth to construct this relationship. The key observation, to me, is Lemma 4.7, which shows a similar result to Lemma 2.1 of [29] with a “surrogate gradient”. In this way, the paper achieves the first small-loss bound for non-smooth functions, which is novel and interesting. For the gradient-variation bound, similar arguments also hold.

Note that, with this observation, one can directly combine it with existing adaptive methods to achieve adaptive bounds. However, it does not mean the proposed methods are less novel.

This paper is generally well-written and easy to follow. The key finding, section 4.3, is easy to understand.


**Weaknesses:**

For exp-concave functions with bounded gradients (e.g., with no-junk bund assumption),  algorithms such as variant of ONS () can get a O(log L^*) bound, while the algorithm in this paper yields a O(root{L^*} + log T) bound, which is sub-optimal. It would be great if the authors can consider how to get a best-of-both-world bound, which enjoys O(root{L^*} + log T) for non-smooth functions, and O(log L^*) for smooth functions.

To obtain he optimal gradient-variation bound, the proposed algorithms need to know V_T in advance. In the appendix, the authors propose an algorithm that can learn the step size adaptively. However, it somewhat lost the good property of the gradient-variation bound when V_T is small.


**Questions:**

N/A

---

> ### Author Rebuttal · Authors · 2023-08-08
>
> Thank you for correctly pointing out our main contribution and emphasizing the novelty in the proposed methods.
> 1. **On achieving optimal small-loss bounds for both smooth and non-smooth functions simultaneously:** This is an interesting direction. Such generalization requires non-trivial work and seems to be not directly related to the topic of this paper. We will consider this as a future research direction.
> 2. **On the gradual variation bound:**
>    - Our algorithm for the gradual variation bound does not need $V_T$ (see Theorem 6.1) and there is no need for learning the learning rate. It is Algorithm 5, which is for the second-order bound, that needs learning the learning rate. As stated in the paper, we have not found a good interpretation of the second-order bound and hence provide the second-order result only in the appendix.
>    - We do not get the statement on losing the good property when $V_T$ is small. We would appreciate it if you could elaborate.

---

> > ### Comment · Reviewer_mCNh · 2023-08-11
> > **Thanks for the reponse**
> >
> > Thank you for the detailed response, and I do not have further questions.
> >
> > For the prior knowledge of V_T, I made a mistake and thought that the step size of Theorem 6.1 is related V_T (but its actually related to V_t). It would be great if the authors can emphasize this point in the revised version after Theorem 6.1.
> >
> > For the first point, I would like to clarify that my **main** point is to discuss the limitation of the current results, instead of suggesting a new direction, so I think it is related to the topic of this paper.

---

> > > ### Author Response · Authors · 2023-08-14
> > > **Thank you for the response**
> > >
> > > Thank you for the reply. We will emphasize that the step size in Theorem 6.1 does not need $V_T$ and discuss the possibility of obtaining a $O ( \log L_T^\star )$ regret bound in the revision.
> > >
> > > If we still have any misunderstandings, then please let us know. Thank you again for your appreciation of this work!

---

### Decision · Program_Chairs · 2023-09-21

**Decision:**

Accept (poster)

**Comment:**

This paper provides new results for the classic online portfolio selection problem. Specifically, the main result is a data-dependent regret bound is better than known bounds when the data are "easy", while having O(sqrt{dT}) regret in the worst case, with O(d) running time per iteration. Crucially, the algorithm doesn't need to assume bounded gradients (also known as the "no-junk-bonds" assumption). Technically, the paper introduces new analysis of optimistic FTRL with self-concordant functions, which are not necessarily barriers. While the worst case regret is quite suboptimal, all known methods which have O(d) running time per iteration have the same regret bound.

Overall, the reviewers appreciated the contributions of this paper, so I am happy to recommend acceptance. Please take the reviewer comments into account while preparing the final version. Specifically, it would be nice if the authors can include a statement for more general losses in their paper. This may help others to build on this work and highlight the techniques developed in the paper.